# Photo-thermal coupling to enhance $CO_2$ hydrogenation toward $CH_4$ over Ru/MnO/$Mn_3O_4$

Jianxin Zhai[1,2], Zhanghui Xia[1,2], Baowen Zhou [3] ✉, Haihong Wu [1,2] ✉, Teng Xue[1,2], Xiao Chen[1,2], Jiapeng Jiao[1,2], Shuaiqiang Jia[1,2], Mingyuan He [1,2] ✉ & Buxing Han [1,2,4] ✉

Upcycling of $CO_2$ into fuels by virtually unlimited solar energy provides an ultimate solution for addressing the substantial challenges of energy crisis and climate change. In this work, we report an efficient nanostructured Ru/MnO$_x$ catalyst composed of well-defined Ru/MnO/$Mn_3O_4$ for photo-thermal catalytic $CO_2$ hydrogenation to $CH_4$, which is the result of a combination of external heating and irradiation. Remarkably, under relatively mild conditions of 200 °C, a considerable $CH_4$ production rate of 166.7 mmol $g^{-1}$ $h^{-1}$ was achieved with a superior selectivity of 99.5% at $CO_2$ conversion of 66.8%. The correlative spectroscopic and theoretical investigations suggest that the yield of $CH_4$ is enhanced by coordinating photon energy with thermal energy to reduce the activation energy of reaction and promote formation of key intermediate COOH* species over the catalyst. This work opens up a new strategy for $CO_2$ hydrogenation toward $CH_4$.

Upcycling of $CO_2$ into fuels with the use of green hydrogen presents a promising route for addressing the challenges of energy crisis and climate change[1–3]. Among a variety of products from $CO_2$ hydrogenation, $CH_4$ is regarded as an ideal energy vector owing to its merits of high energy density, widely available infrastructure of storage, transportation, and utilization[4,5]. To date, a broad range of catalytic systems have been developed for $CO_2$ hydrogenation toward $CH_4$[6]. However, because of the inert nature of $CO_2$ and complex reaction network, efficient production of $CH_4$ from $CO_2$ hydrogenation is challenging, suffering from unsatisfactory activity, harsh reaction condition and extensive thermal input[7–9]. It is imperative to explore new and green methods for the conversion of $CO_2$ toward $CH_4$.

Photo-thermal-catalysis presents a synergistic configuration for mediating chemical reactions by simultaneously taking advantage of charge carriers and thermal energy[10,11]. Thus far, there is a growing number of researches on photo-thermal catalytic $CO_2$ hydrogenation toward $CH_4$ and remarkable progress has been made[12,13]. For example, Liu et al. reported that $Co_7Cu_1Mn_1O_x$ (200) was active for $CH_4$ synthesis from $CO_2$ hydrogenation with an production rate of 14.5 mmol $g^{-1}$ $h^{-1}$ and a selectivity of 85.3% at 200 °C[14]. Zou's group demonstrated that $Ru@Ni_2V_2O_7$ catalyst exhibited $CH_4$ production rate of 114.9 mmol $g^{-1}$ $h^{-1}$ and 99.3% selectivity at 350 °C[15]. Overall, the performance of the catalytic systems is still far away from practical applications and the reaction mechanism remains largely unknown[16]. It is very desirable to explore a strategy for mediating $CO_2$ hydrogenation toward $CH_4$ with high efficiency and selectivity[17].

Among a broad range of $CO_2$ hydrogenation catalysts, Ru-based catalysts exhibit great potential in $CO_2$ hydrogenation toward $CH_4$ because of their unique catalytic properties[18]. Apart from metal centers, the support also plays a critical role in $CO_2$ hydrogenation by

[1]Shanghai Key Laboratory of Green Chemistry and Chemical Processes, State Key Laboratory of Petroleum Molecular & Process Engineering, School of Chemistry and Molecular Engineering, East China Normal University, Shanghai 200062, China. [2]Institute of Eco-Chongming, Shanghai 202162, China. [3]Key Laboratory for Power Machinery and Engineering of Ministry of Education, Research Center for Renewable Synthetic Fuel, School of Mechanical Engineering, Shanghai Jiao Tong University, Shanghai 200240, China. [4]Beijing National Laboratory for Molecular Sciences, CAS Key Laboratory of Colloid and Interface and Thermodynamics, CAS Research/Education Center for Excellence in Molecular Sciences, Institute of Chemistry, Chinese Academy of Sciences, Beijing 100190, China. ✉e-mail: zhoubw@sjtu.edu.cn; hhwu@chem.ecnu.edu.cn; Mingyuanhe@126.com; hanbx@iccas.ac.cn

influencing the geometric and electronic properties of active sites. Particularly, $MnO_x$ is considered a promising support for hydrogenation reactions due to some obvious advantages[19]. It is worth of noting that the multiple valences and reducible effect of $MnO_x$ confers flexible mediation capability on the catalysts[20]. The integration of Ru species with $MnO_x$ is thus highly promising for efficient $CO_2$ hydrogenation toward $CH_4$.

In this work, a nanostructured $Ru/MnO_x$ photo-thermal catalyst composed of well-defined $Ru/MnO/Mn_3O_4$ at reaction temperature was designed and prepared for $CO_2$ hydrogenation toward $CH_4$. A prominent $CO_2$ conversion of 66.8% was achieved with a superior selectivity of 99.5% and a $CH_4$ production rate of 166.7 mmol $g^{-1} h^{-1}$ at relatively mild temperature of 200 °C (normalized by the amount of catalyst (~15 mg)), which is the result of a combination of external heating and irradiation. The correlative spectroscopic characterizations and theoretical calculations revealed that the structural evolution of $Ru/MnO_x$ into well-defined $Ru/MnO/Mn_3O_4$ was facilitated by Ru-mediated H-spillover in $MnO_x$ and the activity was enhanced by the synergistic effects of photon energy and thermal energy via reducing the activation energy of reaction and accelerating the key intermediate of COOH* species formation over the catalyst.

## Results

### Fabrication and characterization of $Ru/MnO_x$

The synthesis process of $Ru/MnO_x$ was schematically shown in Fig. 1a. Typically, $MnO_x$ nanoparticles were first prepared via a straightforward hydrothermal method. Ru sites were then anchored onto $MnO_x$ by photo-deposition under argon atmosphere[21]. The content of Ru in $Ru/MnO_x$ was measured by inductively coupled plasma optical emission spectrometry (ICP-OES) (Supplementary Table 1). If not specifically noted, the content of Ru in $Ru/MnO_x$ was referred to be 7.3 wt%. The morphologies and structures of the synthesized materials were characterized by scanning electron microscopy (SEM) and transmission electron microscopy (TEM). As shown in Figs. 1b, c, $MnO_x$ displayed variable morphologies of hexagonal, octahedral, and square schistose crystals. The morphology of $MnO_x$ did not change considerably after the addition of Ru species and the average size of the deposited Ru nanoclusters is about 1.07 ± 0.26 nm (Fig. 1d and Supplementary Figs. 1, 2). The energy dispersive spectroscopy elemental mapping in Fig. 1e exhibited the even distribution of Mn, O, and Ru, which is indicative of the successful synthesis of $Ru/MnO_x$.

The Rietveld refinement of X-ray powder diffraction (XRD) results in Supplementary Fig. 3 and Supplementary Table 2 indicated that the $MnO_x$ nanoparticles were mainly composed of $Mn_3O_4$ (JCPDS No. 80-0382), $MnO_2$ (JCPDS 72–1806) and MnOOH (JCPDS No. 18-0804)[22–24]. Meanwhile, the XRD pattern of $Ru/MnO_x$ showed that the content of $MnO_2$ phase decreased slightly, indicating that the process of photodeposition of Ru had a slight reduction effect on $MnO_2$. The structure of the samples was further characterized by Fourier transform infrared spectroscopy (FT-IR) spectroscopy (Supplementary Fig. 4). The peaks at 513 and 621 $cm^{-1}$ were attributed to the distortion vibration of Mn-O in octahedral sites and Mn-O stretching modes in tetrahedral sites, respectively[25]. Besides, the typical peaks at 947 and 1074 $cm^{-1}$ were attributed to the vibration of hydroxyl in MnOOH[26]. Meanwhile, Raman spectrometer was employed to study the metal-support interaction between Ru and $MnO_x$. As illustrated in Supplementary Fig. 5,

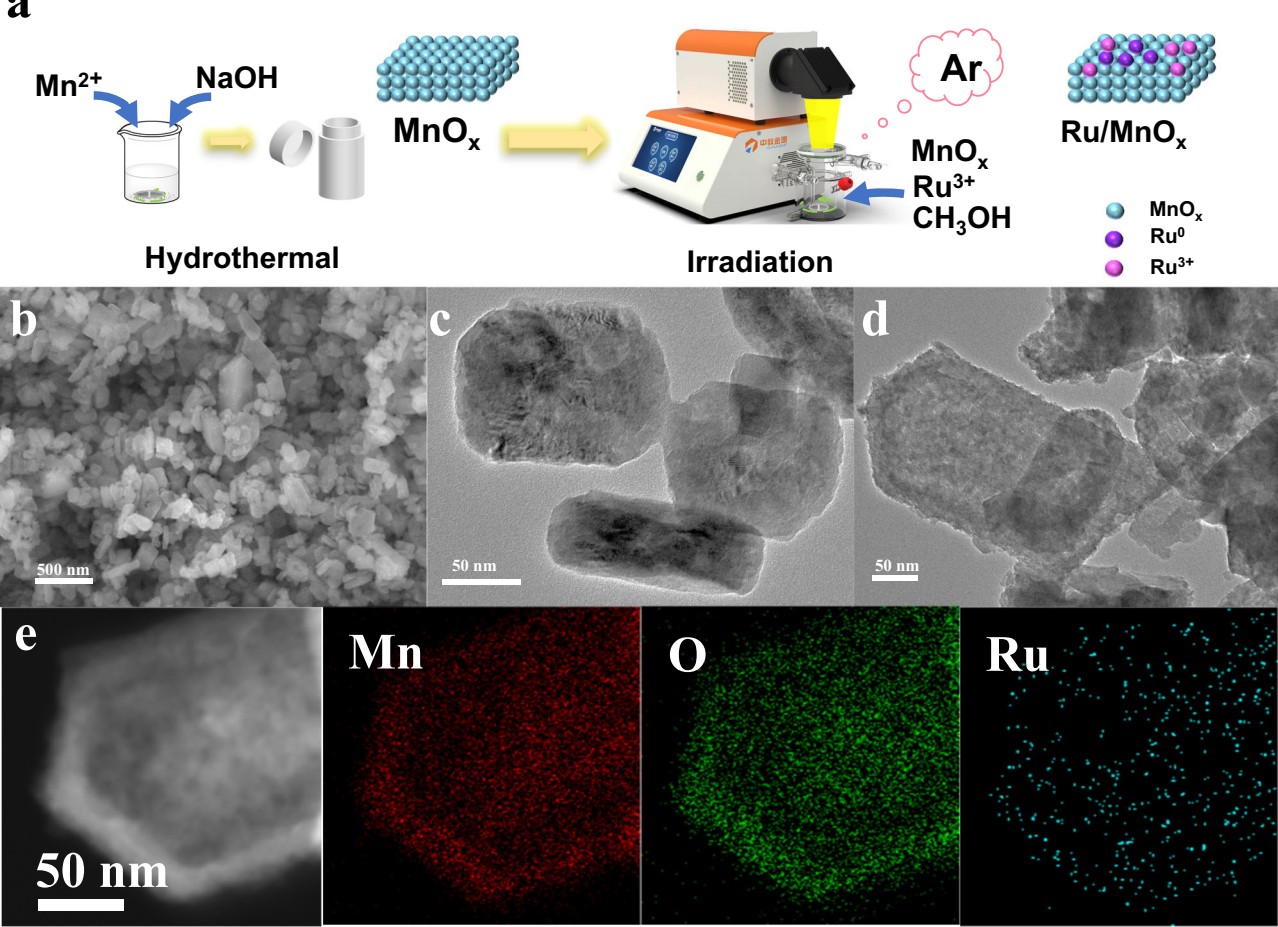

**Fig. 1 | Structural characterization of the catalysts. a** Schematic illustration of the synthesis of $Ru/MnO_x$; **b** SEM image of $MnO_x$; **c, d** TEM image of $MnO_x$ and $Ru/MnO_x$; **e** STEM image and elemental mapping of $Ru/MnO_x$.

compared with the pristine $MnO_x$, the introduction of Ru species led to a blue shift of ~7 wavenumbers, and the main peak at $637\,cm^{-1}$ is assigned to $A_{1g}$ mode of crystalline $Mn_3O_4$, validating the strong metal–support interaction between Ru and $MnO_x$[27]. Moreover, as characterized by $CO_2$ adsorption isotherm and $N_2$ adsorption–desorption isotherms, the addition of Ru species enhanced the $CO_2$ adsorption capacity; and specific surface area of the catalyst was enlarged accordingly (Supplementary Figs. 6, 7). Such improvements are beneficial for the interaction between reactants and mass transfer, thus facilitating $CO_2$ methanation. In addition, X-ray photoelectron spectroscopy (XPS) was examined to gain more insight into surface chemical state and electronic structure of the catalysts (Supplementary Fig. 8). For Ru/$MnO_x$, $Ru^0$ is identified by the peaks observed at ca. 462.7 and 484.8 eV, implying the reduction of Ru species under light irradiation[28]. Additionally, XPS analysis of the Mn species in all tested samples can be deconvolved into $Mn^{4+}$, $Mn^{3+}$, and $Mn^{2+}$[23,27]. Apparently, multiple valences of $MnO_x$ indicate the reducible nature of as-prepared catalyst.

## Photo-thermal catalytic $CO_2$ hydrogenation

The catalytic performance of Ru/$MnO_x$ was evaluated at 200 °C in the batch reactor setup by feeding $CO_2$/$H_2$ mixed gas (the desired temperature was achieved by a combination of external heating and irradiation from the Xe lamp) and $CH_4$ was identified as the dominant product, with no liquid products produced (Supplementary Figs. 9, 10). As shown in Fig. 2a, $MnO_x$ was hardly active for $CO_2$ hydrogenation toward $CH_4$. In contrast, after the addition of Ru species into $MnO_x$, considerable activity for $CH_4$ formation was achieved, indicating that Ru species could serve as effective active site for the reaction. Notably, the catalytic activity of Ru/$MnO_x$ gradually increased with an increasing amount of Ru. At a Ru content of 7.3 wt%, the catalyst displayed a decent $CH_4$ activity of $103.7\,mmol\,g^{-1}\,h^{-1}$. The $CH_4$ activity was increased by further increasing the loading content of Ru, but the trend slowed down. The effect of $CO_2$/$H_2$ ratios on the yield of $CH_4$ was then examined. It was observed that the yield of $CH_4$ monotonically

increased with the $H_2$ proportion in the $H_2$/$CO_2$ mixture (Fig. 2b). A distinct $CH_4$ production rate of $166.7\,mmol\,g^{-1}\,h^{-1}$ was obtained for Ru/$MnO_x$ at a relatively high $H_2$/$CO_2$ ratio of 4/1, highlighting the importance of adequate $H_2$ supply during the reaction to enhance its activity. Furthermore, as shown in Supplementary Fig. 11, we studied the influence of the total pressure on the reaction at high $H_2$/$CO_2$ ratio (4/1). The activity was enhanced markedly with the elevating total pressure, but became slowly when the pressure exceeded 1 MPa.

As shown in Supplementary Fig. 12, the control experiment showed that no carbonaceous products were detected without catalyst or reactant gas, confirming that $CH_4$ was catalytically produced from $CO_2$ hydrogenation. Furthermore, when the photon energy was coupled with external thermal energy, the evolution rate of $CH_4$ was substantially enhanced compared to that achieved by thermo-catalysis in the temperature range examined (Fig. 2c). In addition, when the reaction temperature was kept at 200 °C by an external temperature-controlling system, the $CH_4$ evolution rate of Ru/$MnO_x$ could be further enhanced by increasing the light intensity, reaching $166.7\,mmol\,g^{-1}\,h^{-1}$ at $2.5\,W\,cm^{-2}$ (Fig. 2d). These results provide solid support that the introduction of photon energy can significantly enhance $CO_2$ hydrogenation reaction. Meanwhile, the activation energies of Ru/$MnO_x$ under thermal and photo-thermal conditions were estimated to be 99.9 and $68.5\,kJ\,mol^{-1}$, respectively (Fig. 2e). The decreased activation energy and the corresponding non-parallel plots were indicative of a synergy between photon energy and thermal energy, altering the catalytic mechanism when photons were involved. Moreover, the conversion of $CO_2$ toward $CH_4$ as a function of reaction time is shown in Fig. 2f. It was found that the yield of $CH_4$ continuously increased with the reaction time, suggesting the continuous generation of $CH_4$ from $CO_2$ hydrogenation over Ru/$MnO_x$. Impressively, with a reaction time of 4 h at 200 °C, a considerable $CO_2$ conversion of 66.8% was achieved with a high selectivity of 99.5% with an appreciable $CH_4$ production rate of $166.7\,mmol\,g^{-1}\,h^{-1}$. Furthermore, the photo-thermal catalytic performance of the Ru/$MnO_x$ catalyst was also assessed in a fixed-bed reactor. As shown in Supplementary Figs. 13, 14,

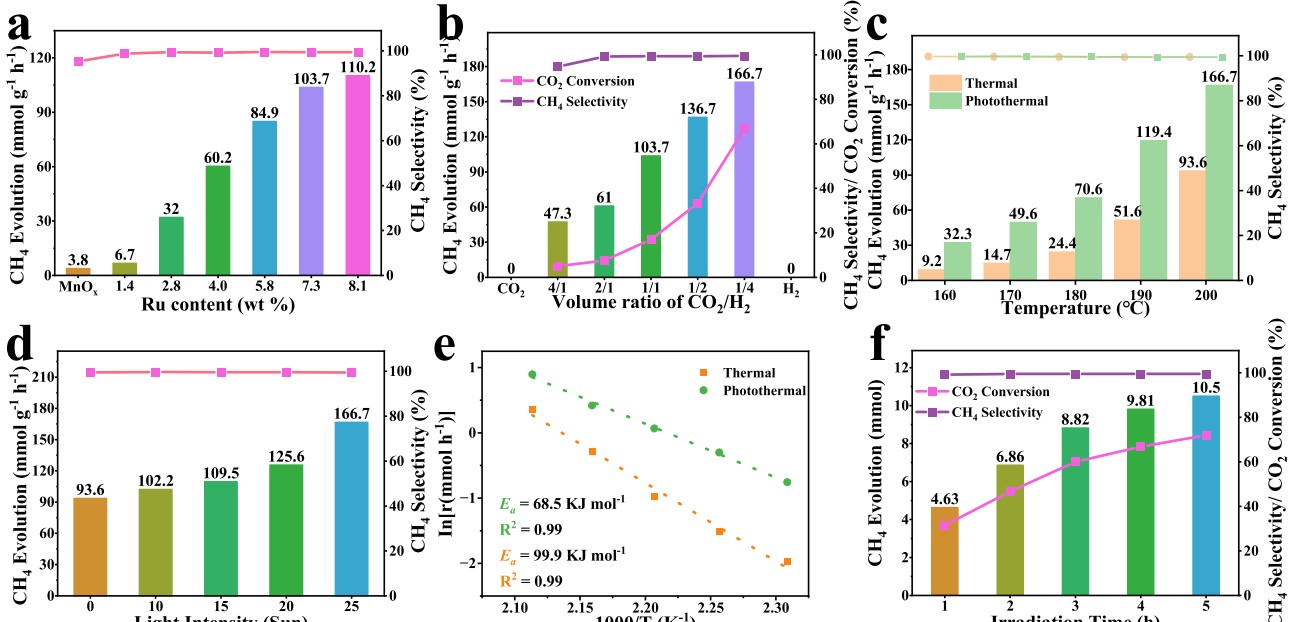

**Fig. 2 | Photo-thermal-catalytic performance. a** Influence of the Ru content on $CH_4$ evolution rate over Ru/$MnO_x$; **b** Influence of $CO_2$/$H_2$ volume ratio in the feedstock on $CH_4$ evolution rate over Ru/$MnO_x$; **c** Temperature-dependent $CH_4$ generation rate over Ru/$MnO_x$ under photo-thermal and thermal conditions; **d** Influence of light intensity on $CH_4$ evolution rate over Ru/$MnO_x$; **e** Corresponding Arrhenius plot with activation energies noted under photo-thermal/thermal conditions over the Ru/$MnO_x$ catalysts. **f** $CH_4$ evolution as a function of reaction time over Ru/$MnO_x$. Reaction conditions: 15 mg of catalyst, full-arc 300 W UV-xenon lamp, $2.5\,W\,cm^{-2}$, 200 °C, irradiation time 4 h, initial pressure 1 MPa ($H_2$/$CO_2$ = 1/1) for Fig. 2a or initial pressure 1 MPa ($H_2$/$CO_2$ = 4/1) for Fig. 2c–f.

at a gas hourly space velocity (GHSV) of 20,000 mL g$^{-1}$ h$^{-1}$, the catalytic activity of Ru/MnO$_x$ gradually increased with an increasing temperature, and its activities under photothermal conditions were higher than those under thermal conditions, which further proves that the involved photons were prone to promote the formation of CH$_4$ in the fixed-bed reactor. Meanwhile, under the conditions of 200 °C and 2.5 W cm$^{-2}$ irradiation, the catalytic activity of Ru/MnO$_x$ remained acceptably stable after 20 h at a high GHSV of 40,000 mL g$^{-1}$ h$^{-1}$ (Supplementary Fig. 15). A CO$_2$ conversion of 29.5% was achieved with an excellent selectivity of 99.5% and a high space-time yield (STY) of 95.8 mmol$_{CH_4}$ g$^{-1}$ h$^{-1}$. The decrease in activity was probably caused by catalyst agglomeration and carbon deposition as confirmed by the TEM and Thermogravimetric-mass spectrometric (TG-MS) of the catalyst after the reaction (Supplementary Figs. 16, 17). In addition, as characterized by XPS, for Ru/MnO$_x$, after the reaction, the composition of Mn$^{4+}$ disappeared, and the peak appearing at 647 eV is the satellite peak of Mn$^{2+}$, demonstrating that the multiphase MnO$_x$ support underwent partial reduction[29]. Ru species were completely converted into Ru$^0$, indicating that the in-situ generated Ru can act as a metal active center to enhance the dissociation of H$_2$ and spillover of hydrogen to the MnO$_x$ support (Supplementary Fig. 18). Therefore, the actual composition of the catalyst during the reaction was altered, and the observations above suggested the H-spillover effects in Ru/MnO$_x$, which will be discussed further in the following.

## Origin of the superior activity over Ru/MnO$_x$

The optical and electronic properties of the catalyst play a vital role in photo-thermal-catalytic activity. Thereby, they were characterized by various spectroscopy techniques. Firstly, the UV–Vis–NIR diffuse reflectance spectra of different samples were measured to study the light absorption capacity. As illustrated in Fig. 3a, MnO$_x$, as an excellent semiconducting support, exhibits suitable light absorption in the UV–Vis region. The light absorption of MnO$_x$ was further improved by the addition of Ru species, having stronger and broader light absorption from UV to NIR wavelength. Meanwhile, due to the broadening of the wavelength range of light absorption, a strong photothermal effect was expected[30,31]. As shown in Supplementary Fig. 19, under 2.5 W cm$^{-2}$ illumination, the measured average temperature of Ru/MnO$_x$ reached 137.9 °C, higher than that of MnO$_x$ (115.4 °C), indicating that both Ru and MnO$_x$ contributed to the photothermal effect. In addition, time-resolved photoluminescence spectroscopy showed that the charge carriers lifetime of bare MnO$_x$ ($\tau_1$ = 1.05 ns, $\tau_2$ = 13.51 ns) was obviously prolonged by the incorporation of Ru species ($\tau_1$ = 1.33 ns, $\tau_2$ = 17.14 ns for Ru/MnO$_x$) in Supplementary Fig. 20[32]. The behavior of charge carriers was further investigated by transient photocurrent spectra, and the results are plotted in Supplementary Fig. 21. Under light irradiation, Ru/MnO$_x$ exhibits a significantly higher photocurrent intensity than MnO$_x$[33]. The results above show that both optical and electronic properties of MnO$_x$ were enhanced by the introduction of Ru species.

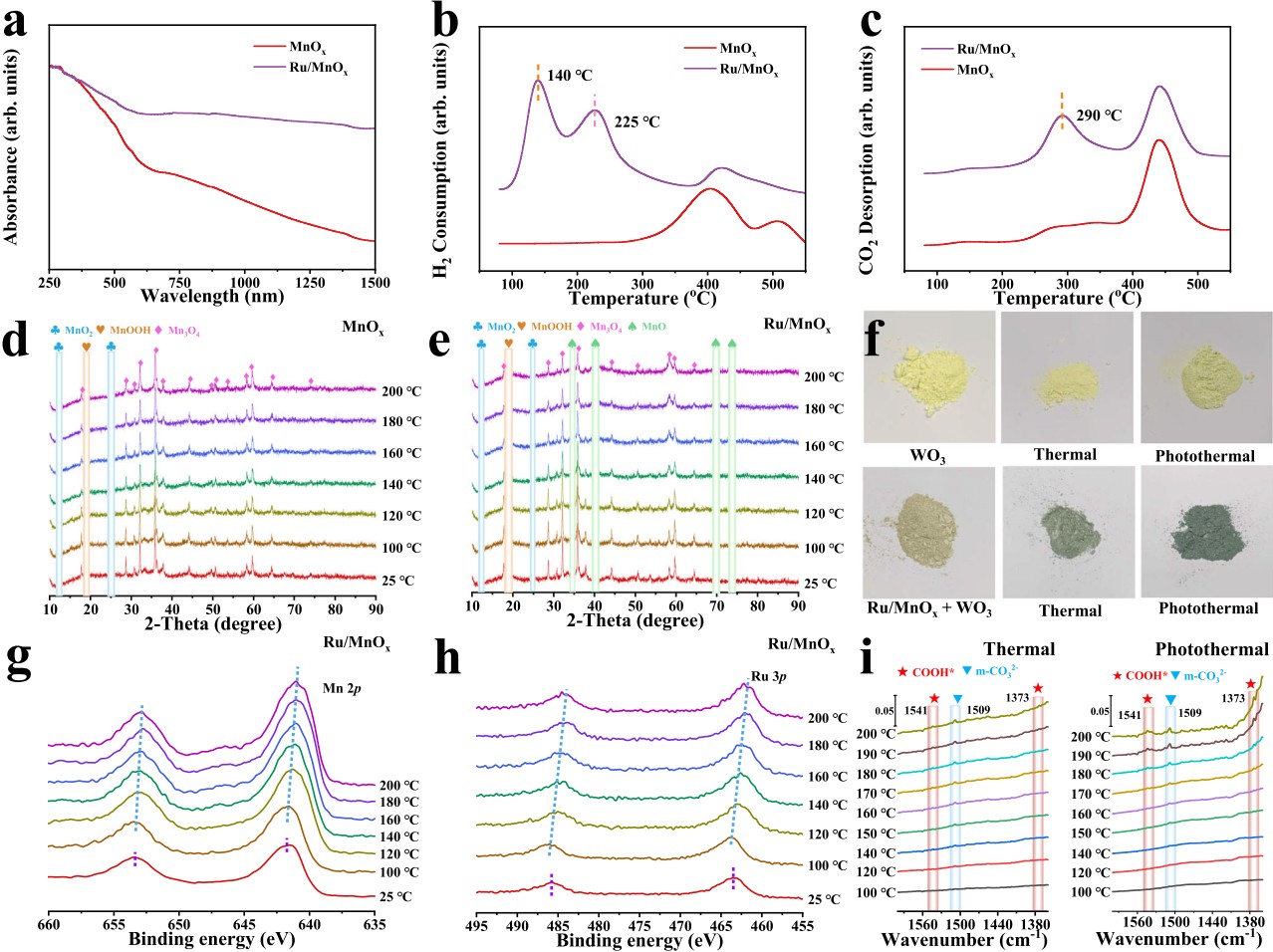

**Fig. 3 | The mechanism analysis. a** UV–Vis–IR absorption spectra of MnO$_x$ and Ru/MnO$_x$; **b** H$_2$-TPR and **c** CO$_2$-TPD characterization for MnO$_x$ and Ru/MnO$_x$; **d, e** The variable-temperature XRD patterns of MnO$_x$ and Ru/MnO$_x$ recorded under 20% CO$_2$/H$_2$ atmosphere at different temperatures; **f** Photographs of WO$_3$ and the mixture of Ru/MnO$_x$ and WO$_3$ samples after treatment with H$_2$ at 80 °C with a light intensity of 0.3 W cm$^{-2}$ for 20 min; **g, h** XPS spectra of Ru/MnO$_x$ in 20% CO$_2$/H$_2$ atmosphere under variable temperature; **i** Spectra of FT-IR study of Ru/MnO$_x$ at different conditions.

Moreover, the flat band potentials of $MnO_x$ were investigated by Mott-Schottky plots. As shown in Supplementary Fig. 22a, $MnO_x$ is confirmed as a p-type semiconductor due to the negative slope. Meanwhile the valence band edge potential was evaluated to be c.a. 0.29 (0.49 eV vs. NHE, $E_{NHE} = E_{Ag/AgCl} + 0.197$), while the flat-band potential is 0.1–0.3 eV lower than the valence band potential in the p-type semiconductor[34]. In addition, the band gap can be estimated to be 1.26 eV for $MnO_x$ (Supplementary Fig. 22b). Consequently, as shown in Supplementary Fig. 23, considering that the work function of Ru is 4.71 eV, the photo-excited electrons can facilely transfer from $MnO_x$ to Ru sites under light irradiation[35]. Together with the unique catalytic properties, the assembled photo-thermal catalyst of $Ru/MnO_x$ was therefore highly efficient and selective for catalyzing $CO_2$ hydrogenation toward $CH_4$.

The characterizations of $H_2$ temperature programmed reduction ($H_2$-TPR) and $CO_2$ temperature programmed desorption ($CO_2$-TPD) were performed to figure out the important role of Ru and $MnO_x$ in $CO_2$ methanation. Compared to pristine $MnO_x$, $Ru/MnO_x$ exhibited remarkably lowered $H_2$ reduction temperature, which was as low as 140 °C (Fig. 3b), suggesting the superior $H_2$ activation capacity of Ru species. The temperature of $H_2$ activation over $Ru/MnO_x$ was even much lower than that for $CO_2$ hydrogenation (200 °C). Hence, $CO_2$ hydrogenation could benefit from low-temperature $H_2$ activation by Ru species under experimental conditions. Moreover, upon the introduction of Ru species, the reduction temperature peaks of $MnO_x$ were observed to emerge at low temperature range of 180–250 °C. Herein, the partial reduction of the support $MnO_x$ was probably associated with the hydrogen dissociation on Ru and its subsequent spillover to $MnO_x$[36]. The $CO_2$ adsorption properties were then evaluated by $CO_2$-TPD. As shown in Fig. 3c, compared to pristine $MnO_x$, $Ru/MnO_x$ showed remarkable desorption peaks between 250 and 350 °C, indicating that Ru species obviously enhanced the adsorption of $CO_2$[37]. Therefore, the dissociated H can further modulate the hydrogenation activity by directly reacting with $CO_2$ adsorbed on Ru sites or $MnO_x$ support, thereby exerting a positive impact on the methanation.

To further explore the Ru-mediated H-spillover effect on the structural composition of $Ru/MnO_x$, qualitative analysis of the crystal phases of $Ru/MnO_x$ during the $CO_2$ methanation process was conducted by variable temperature XRD measurements, with a focus on revealing the structural evolution of $MnO_x$ at different stages. As illustrated in Fig. 3d, e, all the tested samples are primarily composed of $Mn_3O_4$, $MnO_2$, and MnOOH. For $MnO_x$, $MnO_2$ phase can be reduced when the temperature reaches 100 °C. Meanwhile, the transformation of MnOOH to $Mn_3O_4$ phase is observed between 100 and 200 °C. In contrast, the $Mn_3O_4$ phase remains stable at least 4 h at 200 °C (Supplementary Fig. 24). Interestingly, upon the introduction of Ru species, the variable temperature XRD patterns of $Ru/MnO_x$ showed notable changes. The terminal transforming temperature of MnOOH phase markedly decreases from 200 °C to 140 °C. Then, MnO (JCPDS No. 80-0382) with diffraction peaks at 34.7° (111), 40.5° (200), 70.1° (311) and 73.8° (222) were formed when the temperature rises from 140 °C to 200 °C. The observation above is well consistent with the $H_2$-TPR characterization[38]. As a result, the assembled photo-thermal catalyst of $Ru/MnO_x$ was composed of MnO phase and $Mn_3O_4$ phase at 200 °C. The phases of materials could remain stable at least 4 h at 200 °C, implying that the $Ru/MnO_x$ photo-thermal catalyst composed of well-defined $Ru/MnO/Mn_3O_4$ can efficiently and stably catalyze $CO_2$ hydrogenation to $CH_4$ at reaction temperature. Therefore, the phase transformation of multiple-phase $MnO_x$ is clearly indicative of Ru-mediated H-spillover effect. Furthermore, as shown in Supplementary Fig. 25, the catalytic activity of $MnO_x$ supports surpasses that of the other specific manganese oxide alone. It indicates that the H-spillover effect in $Ru/MnO_x$ can effectively transfer dissociated H to the support

due to the multivalent states ($Mn^{2+}/Mn^{3+}/Mn^{4+}$) with varied reducibility, thereby promoting the hydrogenation reaction.

In addition, to investigate the impact of photons on the H-spillover effect under photothermal conditions, we employed $WO_3$ as a means to quantify the extent of H-spillover effect, by which the spillover hydrogen can migrate and readily react with yellow $WO_3$, resulting in a dark coloration[39,40]. The experiment was conducted at 80 °C with a light intensity of 0.3 W $cm^{-2}$ under 1 MPa $H_2$ to ensure that the temperature induced by the photothermal effect remained below the designated temperature (Supplementary Fig. 26). As shown in Fig. 3f, it was revealed that the color of $WO_3$ remained unchanged under both photothermal and thermal conditions. In contrast, the mixture of $Ru/MnO_x$ and $WO_3$ exhibited a darker color under photothermal conditions compared to thermal conditions. This observation suggests that under photothermal catalysis, the irradiation can enhance the H-spillover effect, thereby promoting the subsequent $CO_2$ hydrogenation reaction.

Meanwhile, XPS was performed to gain insight into the composition and chemical valence. As can be observed in Fig. 3g, h and Supplementary Fig. 27, Ru 3p and Mn 2p peaks of $Ru/MnO_x$ displayed a negative shift from 100 °C to 200 °C. Then, the Mn 2p and Ru 3p spectra remain unvaried within 4 h at 200 °C, which is in good agreement with the variable XRD measurements. Supplementary Fig. 28 shows that the binding energies of Ru shift to 462.7 eV, corresponding to $Ru^0$ and the peaks of $Mn^{4+}$ disappeared. The peak at 647 eV is the satellite peak of $Mn^{2+}$. Such binding energy shifts indicate that $Ru^{3+}$ and $Mn^{4+}$ were reduced under the $CO_2/H_2$ mixed gas ($H_2:CO_2 = 4:1$), which is consistent with the $H_2$-TPR characterization.

To better understand the reaction at molecular level and explore the impact of the involved photon on the reaction, $CO_2$ hydrogenation was studied by FT-IR under different conditions (Fig. 3i and Supplementary Figs. 29, 30). For thermocatalysis, the typical peaks of monodentate carbonates ($m-CO_3^{2-}$, 1509 $cm^{-1}$) and $v$(C-H) vibration of $CH_4$ (1305 $cm^{-1}$) were apparently strengthened by increasing the reaction temperature[41,42]. Notably, the intermediate of formate species was observed at 1541 $cm^{-1}$ (COOH*, $v$(OCO)$_{as}$) and 1373 $cm^{-1}$ (COOH*, $v$(OCO)$_s$) when the reaction temperature increased up to 200 °C[43]. In contrast, upon light irradiation, the typical peaks of COOH* species appeared at a lower reaction temperature of 170 °C[44]. This finding further validated the synergy between photon energy and thermal energy on $CO_2$ hydrogenation toward $CH_4$, and COOH* is the most likely key intermediate during either thermo-catalysis or photothermal-catalysis. Hence, the involved photons were prone to accelerate the formation of intermediate species, thus can significantly reduce the activation energy of $CO_2$ hydrogenation reaction and promote the formation of $CH_4$.

To better understand the mechanism of the superior performance, first-principles density functional theory calculations were carried out on the basis of the models of $Ru/Mn_3O_4$ (321) slabs, $Ru/MnO$ (200) slabs and $Ru/Mn_3O_{4-x}$ (321) slabs that simulated the partial reduction of $Mn_3O_4$ by the H-spillover effect (Supplementary Fig. 31)[45]. As shown in Fig. 4 and Supplementary Fig. 32–34, $Ru/Mn_3O_{4-x}$ has a more negative Gibbs free energy ($\Delta G$) than both $Ru/Mn_3O_4$ and $Ru/MnO$ during the adsorption of $CO_2$, indicating a strong adsorption capacity for $CO_2$, which is beneficial for $CO_2$ hydrogenation ($\Delta G = -0.914$ eV, $Ru/MnO$; $\Delta G = -1.475$ eV, $Ru/Mn_3O_4$; $\Delta G = -1.651$ eV, $Ru/Mn_3O_{4-x}$). Afterwards, notable variations for the subsequent $CO_2$ hydrogenation were observed among $Ru/MnO$, $Ru/Mn_3O_4$, and $Ru/Mn_3O_{4-x}$. The formation of COOH* from $CO_2$* is a rate determining step (RDS) for $CO_2$ hydrogenation over $Ru/Mn_3O_{4-x}$ and $Ru/MnO$, which requires 1.232 and 1.544 eV, respectively. The protonation and subsequent dehydration of COOH* results in the generation of the intermediate of CO*, which is the RDS for the $Ru/Mn_3O_4$, ($\Delta G = 1.918$ eV for $Ru/Mn_3O_4$). Notably, compared to HCO* formation, the CO*

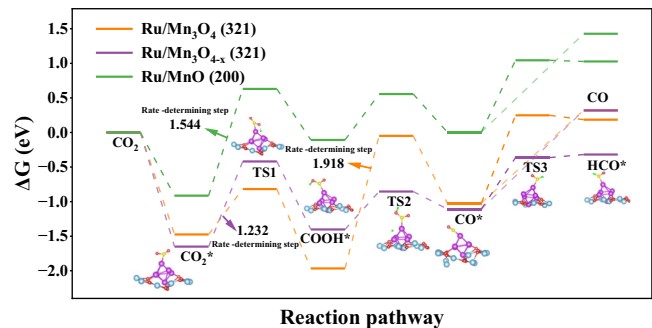

**Fig. 4 | Gibbs free energy pathway for the formation of HCO\* and CO from $CO_2$ over Ru/Mn$_3$O$_4$ (321), Ru/Mn$_3$O$_{4-x}$ (321), and Ru/MnO (200).** The blue, red, purple, yellow, and green spheres represent the Mn, O, Ru, C, and H atoms, respectively, in the calculation model.

desorption from the catalytic surface as CO is relatively difficult for all the samples[46]. As a result, it is favorable to yield CH$_4$ via further hydrogenation. It is worth mentioning that in the process of $CO_2$ hydrogenation, $\Delta G$ of RDS over Ru/Mn$_3$O$_{4-x}$ (1.232 eV) is obviously lower than that on Ru/Mn$_3$O$_4$ ($\Delta G = 1.918$ eV) and Ru/MnO ($\Delta G = 1.544$ eV), thus facilitating the subsequent hydrogenation steps toward CH$_4$. Moreover, as shown in Supplementary Fig. 35, compared with Ru/Mn$_3$O$_{4-x}$ (321) slabs that simulated dark state, the conduction band that simulated the light state moved to the low energy region, indicating that the involved photons were conducive to electron transfer, which is favorable to $CO_2$ hydrogenation toward CH$_4$[47–49]. Together with the FT-IR spectroscopic characterization above, it was rationalized that the synergy between photon energy and thermal energy favored the formation of COOH\*, thus exerting a positive impact on the $CO_2$ methanation over Ru/MnO$_x$, which is generated by the partial reduction of Mn$_3$O$_4$ by Ru-mediated H-spillover effect in $CO_2$ hydrogenation[50].

Based on the spectroscopic and theoretical investigations above, a possible mechanism for photo-thermal-catalytic $CO_2$ hydrogenation over Ru/MnO$_x$ was proposed. The deposited Ru species is highly efficient for H$_2$ dissociation. Benefitting from the strong interaction between Ru and MnO$_x$, Ru-mediated H-spillover effect facilitated the structural evolution of Ru/MnO$_x$ into well-defined Ru/MnO/Mn$_3$O$_4$ at low temperature, and the synergy between photon energy and thermal energy could facilitate further hydrogenation of the adsorbed $CO_2$ molecules over Ru and MnO$_x$ via accelerating the formation of the key intermediate COOH\*. Critically, compared to the process of CO evolution from \*CO, Ru/Mn$_3$O$_{4-x}$ is energetically favored for catalyzing \*CO hydrogenation toward \*HCO, thus promoting the subsequent hydrogenation toward the eventual product of CH$_4$ with high activity and selectivity.

## Discussion

To summarize, a nanostructured Ru/MnO$_x$ photo-thermal catalyst composed of well-defined Ru/MnO/Mn$_3$O$_4$ at reaction temperature was assembled for $CO_2$ hydrogenation toward CH$_4$. The catalyst illustrated a considerable $CO_2$ conversion of 66.8% with a superior selectivity of 99.5% and a CH$_4$ production rate of 166.7 mmol g$^{-1}$ h$^{-1}$ at 200 °C. A series of spectroscopic characterizations and theoretical investigations revealed that benefitting from the strong metal-support interaction between Ru and MnO$_x$, Ru-mediated H-spillover facilitated the structural evolution of Ru/MnO$_x$ into Ru/MnO/Mn$_3$O$_4$ at low temperature, and the synergy between photon energy and thermal energy promoted the yield of CH$_4$ via reducing the activation energy of reaction and facilitating the formation of COOH\* species. This work opens up a new way for photo-thermal-enhanced $CO_2$ hydrogenation toward CH$_4$.

## Methods

### Chemicals

MnSO$_4$·H$_2$O (99.99%), MnO (99.9%), MnO$_2$ (99.95%), Mn$_2$O$_3$ (99.9%), WO$_3$ (99.99%) and Ammonium acetate (99.99%) were purchased from Aladdin Chemical Reagent, Ltd. Acetylacetone (99.5%) was provided by Alfa Aesar Chemical Co. Ltd. H$_2$SO$_4$ (AR), CH$_3$COOH (AR) and CH$_3$OH (99.9%) was purchased from Sinopharm Chemical Reagent Co., Ltd. Ruodium (III) chloride hydrate (99%, Ru 37-40%) and NaOH (96%) were obtained from Beijing InnoChem Science & Technology Co., Ltd. Mn$_3$O$_4$ (99.95%) was obtained from Shanghai Macklin Biochemical Technology Co., Ltd. CO$_2$ (99.995%) and H$_2$(99.9995%) were provided by Air Liquid Houlding Co., Ltd., China. Deionized water was used in all the experiments.

### Materials synthesis

For the synthesis of MnO$_x$, 1 mmol of MnSO$_4$·H$_2$O and 120 mmol of NaOH were dissolved in 30 mL of deionized water with stirring for 0.5 h. The suspension was subsequently transferred into a 100 mL Teflon liner autoclave. The autoclave was heated at 393 K for 12 h, and then cooled to room temperature. The precipitate was washed with distilled water several times until pH = 7, followed by drying in a vacuum at 333 K overnight to obtain the final product.

For the synthesis of Ru/MnO$_x$, 0.1 g MnO$_x$, 10 mL of CH$_3$OH and suitable amount of RuCl$_3$·3H$_2$O (0.02, 0.04, 0.06, 0.08, 0.1, 0.12 mmol) were added in 50 mL of deionized water with stirring in a glass reactor (250 mL) with a quartz window. The chamber was evacuated and then filled with Ar of 1 atm and irradiated with 300 W UV–Xe lamp for 1 h in 3 W cm$^{-2}$. The precipitate was washed with distilled water several times, followed by drying in a vacuum at 333 K overnight to obtain the final product. Unless otherwise specified, Ru/MnO$_x$ indicates that the added amount of RuCl$_3$·3H$_2$O is 0.1 mmol.

Ru/MnO, Ru/Mn$_2$O$_3$, Ru/Mn$_3$O$_4$, and Ru/MnO$_2$ were synthesized through an identical procedure. The major difference was that the corresponding commercial support was added to the reaction. 0.1 g MnO, Mn$_2$O$_3$, Mn$_3$O$_4$ or MnO$_2$, 10 mL of CH$_3$OH, and 0.1 mmol RuCl$_3$·3H$_2$O were added in 50 mL of deionized water with stirring in a glass reactor (250 mL) with a quartz window. The chamber was evacuated and then filled with Ar of 1 atm and irradiated with 300 W UV-Xe lamp for 1 h in 3 W cm$^{-2}$. The precipitate was washed with distilled water several times, followed by drying in a vacuum at 333 K overnight to obtain the final product.

### Photo-thermal CO$_2$ hydrogenation

Photo-thermal $CO_2$ hydrogenation experiments were carried out in stainless steel reactor of 180 mL (CEL-MPR, Beijing China Education Au-Light Co., Ltd.). In a typical experiment, 15 mg of the catalyst powder was dispersed in 10 mL water in reactor, and the reactor was heated at 333 K overnight to volatilize the solvent and the thin film of catalyst was formed for further experiment (the thickness of the catalyst bed was 25.3 ± 4 μm). Prior to photo-thermal reaction, the reactor was sealed and the air in it was substituted with $CO_2$ of 1 MPa three times and then $CO_2$/H$_2$ mixed gas of 1 MPa with desired ratio was charged at room temperature. Then, the external heating and the 300 W UV-Xe lamp (Beijing China Education Au-Light Co., Ltd) with an intensity of 2.5 W cm$^{-2}$ both contributed to maintaining the reactor temperature at 200 °C. After the desired reaction time, the gas products were detected by a gas chromatograph (Agilent GC-8860) and calibrated with a standard gas mixture. To detect liquid products, 10 mL of water was injected into the system after the reaction. Possible liquid products such as methanol, ethanol, acetic acid, and acetaldehyde were detected with an Agilent Technology 7890B gas chromatography system with a flame ionization detector using a DB-WAX-UI column. The possible product formic acid was analyzed by HPLC (Waters 2695) equipped with Aminex HPX-87H column, UV/visible detector (WATER2489), and the mobile phase was 5 mM sulfuric acid

and the flow rate of 0.7 mL/min. The amount of HCHO was analyzed by using the acetylacetone color-development method. Specifically, 1 mL of the as-prepared acetylacetone solution was mixed with 4 mL of the liquid product in a glass bottle, and heated for 5 min in boiling water. The yellow color of the mixed solution was then investigated. Afterwards, a specific amount of solution was taken out, and examined the UV−vis absorption spectrum by using a Shimadzu UV-2700 spectrophotometer. Through the absorbance intensity at 413 nm, the HCHO concentration was obtained. Typically, 100 mL of acetylacetone solution was first prepared by dissolving 15 g of ammonium acetate, 0.3 mL of acetic acid, and 0.2 mL of acetylacetone in water, and was stored in refrigerator with 2−6 °C.

The photothermal $CO_2$ conversion is also performed in the fixed-bed reactor (CEL-GPPCM, Beijing China Education Au-Light Co., Ltd.) at 200 °C. 150 mg of catalyst and $CO_2/H_2$ mixed flow (20 mL min⁻¹/ 80 mL min⁻¹) were used. A 300 W UV-Xe lamp (Beijing China Education Au-Light Co., Ltd) was used as the light source for the reaction (light intensity: 2.5 W cm⁻²). The products in the effluent gas were periodically analyzed by using a gas chromatograph (GC-7920, Beijing China Education Au-Light Co., Ltd.). STY of $CH_4$ ($mol_{CH_4}$ g⁻¹ h⁻¹), was calculated according to the following equation

$$CH_4 STY = \frac{F_{CO_2,in} \times X_{CO_2} \times S_{CH_4}}{W_{cat} \times V_m} \tag{1}$$

where $F_{CO_2}$, it is the volumetric flow rate of $CO_2$, $X_{CO_2}$ is the $CO_2$ conversion, $S_{CH_4}$ is the $CH_4$ selectivity, $W_{cat}$ is the overall mass of catalyst (g), and $V_m$ is the ideal molar volume of $CO_2$ at standard temperature and pressure.

## Materials characterization

The morphology of the samples was characterized by a Zeiss Sigma HD SEM and a JEOL JEM−2100F TEM. The high-angle annular dark-field scanning transmission electron microscope was operated by EM-ARM300F. A Rigaku Ultima VI XRD was employed to record the X-ray diffraction patterns with a scanning speed of 5°/min between 10° and 90°, which was operated at 25 kV and 35 mA with Cu Kα radiation. XPS measurements were operated on AXIS Supraelectron spectrometer with Al Kα radiation. BET measurements were carried out by $N_2$ at −196 °C in a Quadrasorb evo. Fourier transform-infrared spectroscopy was performed using Nicolet NEXUS670. UV−VIS−NIR diffuse reflectance spectra were obtained by a UV−VIS−NIR spectrophotometer (UV-3600 Plus, Shimadzu, Japan). Raman analysis was conducted on a Thermo Scientific DXR2 Smart Raman Spectrometer with a 532 nm laser. The adsorption isotherms of $CO_2$ were determined at 273 K on a BELSORP-max II equipment. A liquid nitrogen-cooled charge-coupled device spectrometer (Princeton Instruments) and a microchannel plate photomultiplier tube (Hamamatsu) combined with time-correlated single photon counting technique (Edinburgh Instruments) were used for photon counting and lifetimes measurements under 375 nm excitation. The Ru contents were quantified by an inductively coupled plasma emission spectrometer (ICP-OES) on an Optima 8300. The $H_2$-TPR were measured on Micromeritics AutoChem II chemisorption analyzer with a TCD detector, the sample was heated to 200 °C at 10 °C min⁻¹ in an He flow (50 mL min⁻¹) and then cooled to 80 °C. Next, the sample was heated to 700 °C at 20 °C min⁻¹ in a 10% $H_2$/He mixed flow (50 mL min⁻¹) atmosphere and the outlet gas was detected by TCD. The $CO_2$-TPD was measured on Micromeritics AutoChem II chemisorption analyzer with a TCD detector. The sample was heated to 200 °C at 10 °C min⁻¹ in an He flow (50 mL min⁻¹) and then cooled to 80 °C. Next, a 10% $CO_2$/He mixed flow (50 mL min⁻¹) was introduced to the catalyst bed for 0.5 h. The sample was then exposed to He (50 mL min⁻¹) for 0.5 h to remove the weakly adsorbed $CO_2$ from the surface. Finally, the sample was heated to 700 °C at 10 °C min⁻¹ in a He atmosphere and the outlet gas was detected by TCD. The

temperature of samples was recorded by an infrared thermal imaging camera (Fotrfic 315, Shanghai Thermal Imaging Technology Co., Ltd.). Considering the limited ability of the reactor window composed of aluminum oxide to penetrate 7−15 μm of infrared light, the image captures the internal temperature of the reactor by quickly removing the reactor window. Variable temperature XRD measurements were collected by an X-ray diffraction patterns (D8 Advance). The samples were heated in $CO_2/H_2$ mixed flow (10 mL min⁻¹/40 mL min⁻¹) from 25 °C to 200 °C. Data collection after maintaining the specified temperature for 5 min. XPS measurements under variable temperature were operated on Thermo EXCALAB 250Xi electron spectrometer with Al Kα radiation. The catalysts were held on the sample holder and activated with illumination in the pretreatment chamber under $CO_2/H_2$ mixed flow (10 mL min⁻¹/40 mL min⁻¹) from 25 °C to 200 °C. The sample was then introduced into the ultrahigh-vacuum chamber for XPS measurement at room temperature after maintaining the specified temperature for 5 min. FT-IR spectra under variable temperature were recorded with a NICOLET iS50 FTIR spectrometer (Thermo SCIENTIFIC, USA) equipped with a high-temperature reaction chamber and a mercury cadmium telluride (MCT) detector at a resolution of 4 cm⁻¹ and 32 scans per spectrum. The background spectrum was scanned after mixture gas ($CO_2$:$H_2$ = 1:4) was introduced. TG-MS analyses were performed on a thermogravimetric analyser (NETZSCH STA449 F3-QMS403D) instrument under air. The catalysts were held on the sample holder and the reactor was sealed until the air in it was substituted with $CO_2/H_2$ mixed flow (10 mL min⁻¹/40 mL min⁻¹). After that the background spectrum was recorded. Upon reaching the desired temperature through simultaneous external heating and irradiation, the system was maintained for 5 min. Subsequently, the light source was removed for FT-IR measurement. The thickness of the catalyst bed in the batch reactor was measured by laser scanning confocal microscopy LEXT OLS5100.

## H-spillover effect detection by $WO_3$

In a typical experiment, a mixture containing 1 g of $WO_3$ and 0.015 g of catalyst was placed in a quartz glass culture dish. Then the quartz glass culture dish was placed in stainless steel reactor of 180 mL (CEL-MPR, Beijing China Education Au-Light Co., Ltd.). Prior to photo-thermal reaction, the reactor was sealed, and the air was replaced by $H_2$ for three times, followed by filling with $H_2$ (1 MPa). Then, the external heating and the 300 W UV-Xe lamp (Beijing China Education Au-Light Co., Ltd) with an intensity of 0.3 W cm⁻² both contributed to maintain the reactor temperature at 80 °C. After the desired reaction time, the color change of the powder samples was recorded.

## The photoelectrochemical (PEC) tests

The photoelectrochemical tests of the samples were carried out on an electrochemical workstation (CHI660e, Chenhua Instrument, Shanghai, China) by using a three- electrode system. The catalyst was drop-coated on clean FTO glass, which was used as a working electrode, while Pt and Ag/AgCl electrodes acted as counter and reference electrodes, respectively. A 300 W Xe lamp (Aulight, Beijing) acted as the light source and all of the electrochemical tests were carried out in 0.1 mol L⁻¹ sodium sulfate solution.

## Computational methods

We have employed the Vienna Ab initio Simulation Package to perform all density functional theory (DFT) calculations. The elemental core and valence electrons were represented by the projector augmented wave method and plane-wave basis functions with a cutoff energy of 400 eV. Generalized gradient approximation with the Perdew−Burke−Ernzerh of (GGA-PBE) exchange-correlation functional was employed in all the calculations. Geometry optimizations were performed with the force convergence smaller than 0.05 eV/Å. The spin-polarization effect was also considered. A climbing image nudged

elastic band method was used to locate the transition states with the same convergence standard. The spin-polarization effect was also considered. The DFT-D3 empirical correction method was employed to describe van der Waals interactions. The DFT + U approach was introduced to treat the highly localized Mn 2p states, using parameters of $U-J = 4$. For FM of $Mn_3O_4$ and MnO, the initial magnetic moments of Mn atoms were set to be +5 μB. Monkhorst-Pack k-points of $1 \times 1 \times 1$ and $2 \times 2 \times 1$ were applied for all the surface calculations of Ru-$Mn_3O_4$ and Ru-MnO. Half atoms at bottom were fixed in all the calculation. The Gibbs free energy was calculated by the following equation: $\Delta G = \Delta E + \Delta E_{ZPE} - T\Delta S$, where the value of $\Delta E$, $\Delta E_{ZPE}$, and $\Delta S$ denotes the changes of DFT energy, the zero-point energy and the entropy at 473.15 K, respectively.

## Data availability

The data supporting the findings of this work are available within the article and its Supplementary Information files. All the data reported in this work are available from the authors. Source data are provided with this paper.

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

## Acknowledgements
The work was supported by the National Key Research and Development Program of China (2023YFA1507901, 2020YFA0710201) (H.H.W.), the National Natural Science Foundation of China (22109095 (B.W.Z.), 21890761 (B.X.H.), 22121002 (B.X.H.)), the Research Funds of Happiness Flower ECNU (2020ST2203) (B.X.H.), Shanghai Pilot Program for Basic Research-Shanghai Jiao Tong University (No. 21TQ1400211) (B.W.Z.).

## Author contributions
J.X.Z., B.W.Z., H.H.W., M.Y.H., and B.X.H. proposed the project, designed the experiments, and wrote the manuscript; J.X.Z. performed the whole experiments; Z.H.X., X.T., X.C., J.P.J., S.Q.J., B.W.Z., H.H.W, M.Y.H., and B.X.H. assisted in analyzing the experimental data; B.W.Z., H.H.W., M.Y.H., and B.X.H. co-supervised the whole project. All authors discussed the results and commented on the manuscript. All the authors contributed to the overall scientific interpretation and wrote the manuscript.

## Competing interests
The authors declare no competing interests.
