## [Peer Review File · Nature Communications]

Photo-thermal coupling to enhance CO₂ hydrogenation toward CH₄ over Ru/MnO/Mn₃O₄REVIEWER COMMENTS

Reviewer #1 (Remarks to the Author):

The current article approaches photothermal methanation of CO₂ over Ru/MnOx catalysts. The results obtained in this work shows high methane production rate (166 mmol g⁻¹h⁻¹), with a remarkably selectivity (>99 %). In addition, this article provides extensive characterization of the catalysts as well as of the metal-free MnOx. In general terms, this article is clearly written, and easy to read, although some parts may require some rewriting for sake of clarity. Nevertheless, in my opinion, there are important aspects of this work that cast doubts about the soundness of the experimental results, whereas some interpretation of the characterization appears to be flawed. In particular, the following points require further justification and elaboration:

- In the first section regarding characterization of the catalysts, it is stated that Ru does not modify the XRD pattern. However, Fig S2 shows clearly that following Ru addition the low angle reflection at about 12° as well as a smaller one around 25 ° almost disappear. Then, during Ru incorporation some structural changes occur. Furthermore, a clear identification of phases should be given, and ideally the XRD pattern should be analyzed by Rietveld method to ensure that all contributions are accounted.
- Although the methanation activity reported here is notable, better results have been reported recently (see 10.1038/s41929-023-00970-z) using Au/Ce_{0.95}Ru_{0.05}O₂ as catalyst and working in continuous-flow reactor. In contrast, in the present case a batch reactor with very low ratio catalyst mass/volume (15 mg/180 mL) is used. This configuration is far from ideal because gas diffusion and water vapor accumulation is likely to have a significantly influence on the measured activity and, therefore, isolating the real contribution of the catalyst can be difficult. In addition, other experimental details of the catalytic tests require further clarification:
 - o Experimental section does not clarify if heating is achieving exclusively by irradiation with the Xe lamp or if it requires an additional heating system. This is key aspect for understanding the activity tests, particularly the blank experiments of Fig S8 and those of Fig 2D.
 - o As the stoichiometry of the Sabatier reaction requires a CO₂/H₂ ratio of 1:2 it is surprising the authors decided to test lower concentration of H₂ that can favor reverse water gas shift reaction. What is the rationale for testing those conditions? How is the selectivity to methane affected?
 - o Catalytic tests are performed under pressure. This surely promotes methane formation, but the authors should briefly justify the selection of these conditions with regards to other works in the literature.
- Since under operation conditions the real composition of the catalyst is Ru/MnO/Mn₃O₄ it would be rather informative to test catalysts with composition Ru/MnO and Ru/Mn₃O₄, as to ascertain the possible role of these Mn oxides phases.
- The assignment of the different components obtained by deconvolution of XPS, as it displayed in Fig S7 and S9, should be given in these graphs for quick reference.
- It is not clear what are exactly “semi-in situ” conditions for XPS and FTIR analyses. In any case, for these

last experiments as presented in Fig 3H and 3I, they appear to be dominated by gas phase contribution, providing little information about surface species. In fact, the band at 1305 cm^{-1} is very likely due to gas phase methane. Furthermore, I think that in order to show clear indications of the presence of formate, background should be subtracted, and the relevant spectral range should be zoomed.

- DFT analysis show that methane formation is favored on $\text{Ru/Mn}_3\text{O}_4$ with some oxygen vacancies, but these calculations should be compared with the case of MnO , as this phase is also presented in the working catalyst. In addition, if I understand correctly these calculations consider only thermal processes, and therefore the role of photonic activation is not clearly considered in these calculations.

Due to these reasons, I cannot recommend this paper to be published in Nature Communication. Nevertheless, I encourage the author to revise this contribution considering the above-mentioned issues and resubmit an enhanced version to a more specialized journal.

Reviewer #2 (Remarks to the Author):

In this manuscript, the authors synthesized a new photothermal catalyst $\text{Ru/MnO/Mn}_3\text{O}_4$ to convert $\text{CO}_2 + \text{H}_2$ into CH_4 , with a superior selectivity 99.5% and a CO_2 conversion rate of 66.8%. The author characterized the structure of catalysts by XRD, XPS, FTIR, and Raman. The reaction results were analyzed under the influence of different factors such as Ru content, $\text{CO}_2:\text{H}_2$ ratio, temperature, light intensity, and irradiation time. They proposed some reaction pathways by DFT calculations, together with the experimental evidence from FTIR. Interestingly, they observed the Ru-mediated H-spillover effect and found the formation of COOH^* is easier on $\text{Ru/Mn}_3\text{O}_4$, by both experiment and calculation. This work is surely of interests to the community of photothermal catalytic conversion of CO_2 . I would suggest the manuscript to be accepted to the journal of Nature Communications, under the condition that the authors completed the following minor revisions.

1. Could the authors provide the specific morphology and size of Ru particles on the support? For example, are they nanoparticles, atomic cluster, or single atoms?
2. Please clarify in the manuscript what is your definition of photo-thermal catalysis. By “photo-thermal” catalysts, do you mean traditional photocatalysis under the condition of external heating? If so, this might be an “extended” definition of photothermal catalyst, which needs to be further clarified and proved in the manuscript. For photothermal catalysts defined in the literature [see Chem Catalysis, 1, 52-83, 2022; Chem Catalysis, 1, 272-297, 2021], photothermal effect (e.g., photon energy is converted to heat) is the feature of photothermal catalysts. If the $\text{Ru/MnO/Mn}_3\text{O}_4$ catalyst has photothermal effect, which component (e.g., MnO_x , Ru, or both) makes the major contribution to the photothermal effect? To answer this question, the authors should show the maximum temperatures can be reached by the irradiation of Ru/MnO_x and MnO_x , respectively, with increasing irradiation time.
3. In Figure 2C, how do you control the temperature of photothermal catalyst? In principle, reaching a

steady (or maximum) temperature of photothermal catalyst depends on the nature of photothermal effect of materials. If external heating is used to control the photothermal temperature, how may that influence the photothermal effect? In another word, will the photothermal effect of catalysts be influenced by the condition of external heating?

4. On Line 111, which vibrational peak is the blue shift of 7 cm⁻¹ relative to?

5. For the proposed reaction pathways in Figure 4, please comment on which step(s) can be the rate determining step (RDS) for the overall reaction? Energetically, the formation of HCO* seems to be the RDS, the diagram is lack of transition states. Is the formation of COOH* the RDS? If not, why do you think that the difference in COOH* formation between Ru/Mo₃O₄ and Ru/Mo₃O_{4-x} can cause significant difference in their overall reaction kinetics? If possible, the authors should provide the transition search results.

6. Please clarify if the Ru-mediated H-spillover effect in thermal catalysis remains the same under the photothermal condition. Can the light irradiation influence the H-spillover effect, why or why not?

Reviewer #3 (Remarks to the Author):

In this work, Zha and collaborators present a new catalyst based on Ru sites supported on MnO_x for the low-temperature photo-thermal methanation of CO₂. The as-prepared catalyst displayed a remarkable catalytic activity and CH₄ selectivity under reaction conditions owing to the synergy between thermal and non-thermal contributions of light. Mechanistic studies indicate a decrease in the apparent activation energy and an enhancement in the formation of COOH* intermediates under illumination, thus favoring the methanation reaction.

The production of solar fuels and chemicals using CO₂ as feedstock has raised as an interesting alternative to both tackle carbon dioxide emissions and energy crisis. In this context, photo-thermal catalysis overcomes the limitations of traditional photo-catalysis by synergistically combining thermal and non-thermal contributions of sunlight, thus becoming a very dynamic and promising field of research. The results presented by Zha and collaborators seem reassuring, however, given the vast amount of works on photo-thermal catalysis for CO₂ methanation using Ru-based catalyst, I cannot perceive any significant advance in the field. In addition to this, the role of light and heat in the overall reaction mechanism has not been completely discussed and this can lead to misinterpretations in the overall reaction pathway. Furthermore, authors did not provide any stability test to evaluate the long-term activity of the catalyst under reaction conditions.

For these reasons, I cannot recommend the publication of this work in Nature Communications in the present form. Detailed comments to support this decision and suggestions to improve the quality of this work can be found below:

- 1) Authors should provide an analysis on the particle size distribution of Ru on the surface of MnOx. From the available images it is impossible to have an idea of the size of the Ru particles.
- 2) Authors report remarkable methane productions in the order of hundreds of mmol g⁻¹ h⁻¹. Are these catalytic rates normalized by the amount of catalyst (15 mg) or the total amount of Ru present in the sample?
- 3) When it comes to the photo-thermal experiments, could the authors explain in detail the position of the thermocouple in the setup? Is it in contact with the catalyst bed or inserted in the reactor wall? Imprecise temperature measurements can lead to misinterpretations in the contributions of photon and thermal energy to the overall catalytic performance, for instance, in the calculation of apparent activation energy.
- 4) In Fig. 2D, authors studied the effect of the light intensity on the catalytic activity. Was the temperature constant at 200 °C throughout all the intensities? Do the authors attribute the improvement in the performance only to pure non-thermal effects? It is hard to imagine a scenario in which the temperature of the catalyst does not increase upon increasing light intensity, specially taking into account its broad light absorption across the visible and infrared.
- 5) In Fig 2F, authors represented the CH₄ production as a function of the irradiation time. Why did the authors stop the experiment after 4 hours? Longer reaction times would show if higher conversions are achievable.
- 6) Results show a very high methane selectivity in most of the experiments assuming a total reaction time of 4 hours. What happens at shorter reaction times? Is still CH₄ the main product?
- 7) Authors did not provide any stability test of the catalyst, so it is not possible to assess if the material is stable upon consecutive reuses. This type of study is vital to evaluate the practical application of the catalyst, so I encourage authors to perform a series of (at least) five consecutive runs to study the catalyst recyclability.
- 8) Both steady-state and time-resolved PL suggest a charge transfer from MnOx to Ru sites under irradiation. Is this electronic transfer thermodynamically favored? Could the authors provide a band diagram showing the corresponding potentials of MnOx and metallic Ru? Furthermore, authors did not clarify the specific role of these electrons in the overall reaction pathway.
- 9) In Table S2 in SI, please include the amount of Ru in all the samples. For a fair comparison and to avoid misleading conclusions, results should clearly indicate that the methane production rate has been normalized per total mass of catalyst or total mass of Ru. TOF calculations should be also included in the table.

Response Letter

Manuscript ID: NCOMMS-23-30118

Title: “Photo-thermal coupling to enhance CO₂ hydrogenation toward CH₄ over Ru/MnO/Mn₃O₄”

We are very grateful to the referees for the critical comments and the constructive suggestions, which helped us to improve the quality of the manuscript. We have carefully responded to all the questions point-by-point, and have revised the manuscript thoroughly. The changes have been highlighted by yellow background in the revised manuscript.

Reviewer #1 (Remarks to the Author):

The current article approaches photothermal methanation of CO₂ over Ru/MnO_x catalysts. The results obtained in this work shows high methane production rate (166 mmol g⁻¹ h⁻¹), with a remarkably selectivity (>99 %). In addition, this article provides extensive characterization of the catalysts as well as of the metal-free MnO_x. In general terms, this article is clearly written, and easy to read, although some parts may require some rewriting for sake of clarity. Nevertheless, in my opinion, there are important aspects of this work that cast doubts about the soundness of the experimental results, whereas some interpretation of the characterization appears to be flawed. In particular, the following points require further justification and elaboration:
Response: We thank the referee for the encouraging comments, and we have addressed the critical questions and concerns by the referee thoroughly.

Comment 1. In the first section regarding characterization of the catalysts, it is stated that Ru does not modify the XRD pattern. However, Fig S2 shows clearly that following Ru addition the low angle reflection at about 12° as well as a smaller one around 25 ° almost disappear. Then, during Ru incorporation some structural changes occur. Furthermore, a clear identification of phases should be given, and ideally the XRD pattern should be analyzed by Rietveld method to ensure that all contributions are accounted.

Response 1: We thank the referee very much for bringing this question to our attention. Based on the referee’s comment, we have carefully reevaluated our data, and it is found that the decrease in diffraction peaks corresponds to the MnO₂ phase. Furthermore, we have also performed simulations on the XRD pattern by using Rietveld method, which support the notion that the decrease in diffraction peaks is attributed to a reduction in the MnO₂ phase content.

These changes may occur during the mild reduction of the MnO₂ in the photo-deposition process. Regarding the identification of phases and the analysis of the XRD pattern, we acknowledge the importance of providing a clear identification of phases. In the revised manuscript, the discussion has been updated as following: “The Rietveld refinement of X-ray powder diffraction (XRD) results in **Supplementary Fig. 3 and Supplementary Table 2** indicated that the MnO_x nanoparticles were mainly composed of Mn₃O₄ (JCPDS No. 80-0382), MnO₂ (JCPDS 72–1806) and MnOOH (JCPDS No. 18-0804). Meanwhile, the XRD pattern of Ru/MnO_x showed that the content of MnO₂ phase decreased slightly, indicating that the process of photo-deposition of Ru had a slight reduction effect on MnO₂.” (Please see Page 4 in the revised manuscript)

Supplementary Fig. 3 Rietveld refinement result of XRD patterns: (a) MnO_x; (b) Ru/MnO_x.

Supplementary Table 2 Crystal parameters and reliability factors of the refinement for MnO_x and Ru/MnO_x.

Sample	MnO _x			Ru/MnO _x		
Phase	Mn ₃ O ₄	MnOOH	MnO ₂	Mn ₃ O ₄	MnOOH	MnO ₂
Abundance (%)	69.023	25.976	5.002	72.528	27.544	0.928
Space group	I41/amd	P-3m1	C12/m1	I41/amd	P-3m1	C12/m1
a (Å)	5.7702(4)	3.2031(16)	5.1657(61)	5.7698(2)	3.2016(17)	5.1657(61)
b(Å)	5.7702(4)	3.2031(16)	2.8645(61)	5.7698(2)	3.2016(17)	2.8645(33)
c(Å)	9.4544(9)	4.6199(9)	7.0860(32)	9.4490(5)	4.6141(7)	7.0860(32)
Volume(Å ³)	314.796(62)	41.050(42)	104.17(18)	314.569(31)	40.959(44)	104.17(18)
R _{wp}		1.62%			1.71%	
R _p		1.25%			1.33%	
GOF		1.34			1.33	

Comment 2. Although the methanation activity reported here is notable, better results have been reported recently (see 10.1038/s41929-023-00970-z) using Au/Ce_{0.95}Ru_{0.05}O₂ as catalyst and working in continuous-flow reactor. In contrast, in the present case a batch reactor with very low ratio catalyst mass/volume (15 mg/180 mL) is used. This configuration is far from ideal because

gas diffusion and water vapor accumulation is likely to have a significantly influence on the measured activity and, therefore, isolating the real contribution of the catalyst can be difficult. In addition, other experimental details of the catalytic tests require further clarification: Experimental section does not clarify if heating is achieving exclusively by irradiation with the Xe lamp or if it requires an additional heating system. This is key aspect for understanding the activity tests, particularly the blank experiments of Fig S8 and those of Fig 2D. As the stoichiometry of the Sabatier reaction requires a CO₂/H₂ ratio of 1:2 it is surprising the authors decided to test lower concentration of H₂ that can favor reverse water gas shift reaction. What is the rationale for testing those conditions? How is the selectivity to methane affected? Catalytic tests are performed under pressure. This surely promotes methane formation, but the authors should briefly justify the selection of these conditions with regards to other works in the literature.

Response 2: We thank the referee again for the critical comment.

1) Firstly, we appreciate the referees' attention to recent advancements in the field, specifically the work published in "10.1038/s41929-023-00970-z," which demonstrates improved results using Au/Ce_{0.95}Ru_{0.05}O₂ catalyst in a continuous-flow reactor.¹ It is an insightful research and has been included in the revised manuscript as Ref. 12.

2) Secondly, we highly agree with the referee that the limitations of batch reactor setup and the factors of gas diffusion and water vapor accumulation have potential influence on the observed activity. However, despite these limitations, it is believed that this study still provides valuable insights into the field of photothermal methanation of CO₂ research. The highlight of this study was to investigate an efficient catalyst, which allowed us to explore certain aspects of the reaction mechanism and open up a new strategy for photo-thermal CO₂ hydrogenation toward CH₄. Meanwhile, as reported by He and co-workers (Green Chem., 2021, 23, 5775), Liu and co-workers (Adv. Energy Mater., 2022, 12, 2201009) and Zhong and co-workers (Nature catalysis., 2023, 6, 519-530) etc, the explored catalysts that demonstrated good results in the batch reactor setup can also show good performance in continuous-flow setup^{1,2,3}. According to the comment, the photothermal catalytic performance of the Ru/MnO_x catalyst was also assessed in a fixed-bed reactor, and achieved excellent results. It further shows that our strategy is viable. As illustrated in **Supplementary Fig. 13-14**, under the conditions of 200 °C and 2.5 W cm⁻² irradiation, the catalytic activity of Ru/MnO_x remained stable after 20 hours at a high gas hourly space velocity (GHSV) of 40000 mL g⁻¹ h⁻¹. A CO₂ conversion of 29.5% was achieved with an excellent selectivity of 99.5% and a high space time yield (STY) of 95.8 mmol_{CH₄} g⁻¹ h⁻¹. According to the reviewer's suggestion, in the revised manuscript, we have elaborated the description as following, including the experimental setup: "Furthermore, the photothermal catalytic performance of the Ru/MnO_x catalyst was also assessed in a fixed-bed reactor. As illustrated in **Supplementary Fig. 13-14**, under the conditions of 200 °C and 2.5 W cm⁻² irradiation, the catalytic activity of Ru/MnO_x remained stable after 20 hours at a high gas hourly

space velocity (GHSV) of 40000 mL g⁻¹ h⁻¹. A CO₂ conversion of 29.5% was achieved with an excellent selectivity of 99.5% and a high space time yield (STY) of 95.8 mmol_{CH₄} g⁻¹ h⁻¹.” and “The photothermal CO₂ conversion are also performed in the fixed-bed reactor (CEL-GPPCM, Beijing China Education Au-Light Co., Ltd.) at 200 °C. 150 mg of catalyst and CO₂/H₂ mixed flow (20 mL min⁻¹/80 mL min⁻¹) were used. A 300W UV-Xe lamp (Beijing China Education Au-Light Co., Ltd) was used as the light source for the reaction (light intensity: 2.5 W cm⁻²). The products in the effluent gas were periodically analyzed by using a gas chromatograph (GC-7920, Beijing China Education Au-Light Co., Ltd.). STY of CH₄ (mol_{CH₄} g⁻¹ h⁻¹), was calculated according to the following equation

$$\text{CH}_4 \text{ STY} = \frac{F_{\text{CO}_2, \text{in}} \times X_{\text{CO}_2} \times S_{\text{CH}_4}}{W_{\text{cat}} \times V_m}$$

where $F_{\text{CO}_2, \text{in}}$ is the volumetric flow rate of CO₂, X_{CO_2} is the CO₂ conversion, S_{CH_4} is the CH₄ selectivity, W_{cat} is the overall mass of catalyst (g), and V_m is the ideal molar volume of CO₂ at standard temperature and pressure.” (Please see Page 7 and Page 15 in the revised manuscript).

Supplementary Fig. 13 The images of (a) the photo-thermal catalytic performance evaluation process carried out in the flow reaction system and (b) the fixed-bed quartz tube reactor.

Supplementary Fig. 14 The photothermal catalytic performance of Ru/MnO_x catalyst in a fixed-bed reactor. Reaction conditions: 150 mg of catalyst, full-arc 300 W UV-xenon lamp, 2.5 W cm⁻², 200 °C, initial pressure 0.1 MPa, CO₂/H₂ mixed flow (20 mL min⁻¹/80 mL min⁻¹).

3) Thirdly, we would like to clarify that the temperature of 200 °C was achieved by the combined effect of external heating and irradiation from the Xe lamp. As illustrated in **Supplementary Fig. 9**, in order to ensure accurate temperature measurement and to maintain uniform temperature throughout the entire reaction system, the thermocouple was positioned at a distance of 1 cm above the catalyst, in the middle of the reactor. This was done to avoid any contact between the thermocouple and the bottom of the reactor, which could result in inaccurate temperature readings. By doing so, the temperature measurement can accurately reflect the temperature of the entire catalytic reaction system. To provide a clear understanding of the experimental conditions and avoid any misleading information, in the revised manuscript, we have elaborated the description as following: “The catalytic performance of Ru/MnO_x was evaluated at 200 °C in the batch reactor setup by feeding CO₂/H₂ mixed gas (the desired temperature was achieved by a combination of external heating and irradiation from the Xe lamp) and CH₄ was identified as the dominant products, with no liquid products produced (**Supplementary Fig. 9**)” and “Then, the external heating and the 300W UV-Xe lamp (Beijing China Education Au-Light Co., Ltd) with an intensity of 2.5 W cm⁻² were both contributed to maintain the reactor temperature at 200 °C.” (Please see Page 5 and Page 15 in the revised manuscript).

Supplementary Fig. 9 (a) Photograph of the apparatus setup for photo-thermal CO₂ experiments in the batch reactor; (b) Schematic illustration of the photo-thermal reactor.

4) Fourthly, with respect to the CO₂/H₂ stoichiometry, we agree that the Sabatier reaction typically requires a ratio greater than 1:2. However, we intentionally selected a lower H₂ concentration (CO₂/H₂ ratio is 1/1) when testing the catalytic activity of MnO_x with varying Ru contents, with the aim of not only to investigate the optimal Ru loading, but also to explore Ru's impact on the selectivity towards CH₄. As shown in **Fig. 2a**, the results demonstrate that even at lower Ru loadings, the catalysts consistently exhibit exceptional CH₄ selectivity, emphasizing Ru's outstanding methanation capabilities. Furthermore, as shown in **Fig. 2b**, at a Ru content of 7.3 wt%, the Ru/MnO_x catalyst still exhibits 94.7% CH₄ selectivity even at a lower H₂

concentration (CO₂/H₂ ratio is 4/1), further highlighting the superior methanation performance of the Ru/MnO_x catalyst.

5) Fifthly, on the basis of the referee's comment, we have investigated the catalytic performance at different total pressure on CH₄ evolution. It was validated that 1 MPa was the optimal total pressure and the catalyst displayed a decent CH₄ activity of 166.7 mmol g⁻¹ h⁻¹. By further increasing the total pressure, the CH₄ activity increased, but slowly. For better reading, the following statement has been added in the revised manuscript: "Furthermore, as shown in **Supplementary Fig. 11**, we studied the influence of the total pressure on the reaction at high H₂/CO₂ ratio (4/1). The activity was enhanced markedly with the elevating total pressure, but became slowly when the pressure exceeded 1 MPa." (Please see Page 6 in the revised manuscript)

Supplementary Fig. 11 Influence of total pressure on CH₄ evolution rate over Ru/MnO_x; Reaction conditions: 15 mg of catalyst, full-arc 300 W UV-xenon lamp, 2.5 W cm⁻², 200 °C, irradiation time 4 hours, H₂/CO₂=4/1.

Comment 3. Since under operation conditions the real composition of the catalyst is Ru/MnO/Mn₃O₄ it would be rather informative to test catalysts with composition Ru/MnO and Ru/Mn₃O₄, as to ascertain the possible role of these Mn oxides phases.

Response 3: We thank the referee very much for bringing this important question to our attention. On the basis of the referee's comment, we have investigated the catalytic properties of different manganese oxide supports for CH₄ evolution. As shown in **Supplementary Fig. 22**, all the tested manganese oxide supports exhibit certain catalytic activity. Notably, the commercially available Mn₃O₄ support outperforms other commercially available supports like MnO₂, Mn₂O₃ and MnO, indicating the significance of the Mn₃O₄ phase in the reaction. Furthermore, due to the advantages of multiple valences (Mn²⁺/Mn³⁺/Mn⁴⁺) and reducible effect, the MnO_x support exhibits the highest catalytic activity, further indicating that Ru-mediated H-spillover effect on

the MnO_x can efficiently transfer dissociated H to the support, thereby promoting the hydrogenation reaction.^{4,5} According to the reviewer's suggestion, in the revised manuscript, we have devoted one paragraph to elaborate the data: "Furthermore, as shown in **Supplementary Fig. 22**, the catalytic activity of MnO_x supports surpasses that of the other specific manganese oxide alone. It indicates that the H-spillover effect in Ru/MnO_x can effectively transfer dissociated H to the support due to the multivalent states ($\text{Mn}^{2+}/\text{Mn}^{3+}/\text{Mn}^{4+}$) with varied reducibility, thereby promoting the hydrogenation reaction." (Please see Page 10 in the revised manuscript).

Supplementary Fig. 22 Influence of various manganese oxide on CH_4 evolution rate. Reaction conditions: 15 mg of catalyst, full-arc 300 W UV-xenon lamp, 2.5 W cm^{-2} , $200 \text{ }^\circ\text{C}$, irradiation time 4 hours, initial pressure 1 MPa ($\text{H}_2/\text{CO}_2=1/1$).

Comment 4. The assignment of the different components obtained by deconvolution of XPS, as it displayed in Fig S7 and S9, should be given in these graphs for quick reference.

Response 4: We thank the referee again. Based on the referee's suggestion, the assignment of the different components obtained by deconvolution of XPS was conducted and the results are shown in **Supplementary Fig. 8**, **Supplementary Fig. 15** and **Supplementary Fig. 25** for quick reference as suggested by the referee.

Supplementary Fig. 8 (a-b) High-resolution Mn 2p XPS spectra of MnO_x and Ru/MnO_x; (c) High-resolution Ru 3p XPS spectra of Ru/MnO_x; (d) XPS survey spectrum of Ru/MnO_x.

Supplementary Fig. 15 XPS spectra of Ru/MnO_x after reaction in 4 h at 200 °C in the batch reactor: (a) High-resolution of Mn 2p XPS spectra; (b) High-resolution of Ru 3p XPS spectra.

Supplementary Fig. 25 Semi *in-situ* XPS spectra of Ru/MnO_x after reacting at 200 °C for 4 h in a 20% CO₂/H₂ atmosphere: (a) High-resolution of Mn 2p XPS spectra; (b) High-resolution of Ru 3p XPS spectra.

Comment 5. It is not clear what are exactly “semi-in situ” conditions for XPS and FTIR analyses. In any case, for these last experiments as presented in Fig 3H and 3I, they appear to be dominated by gas phase contribution, providing little information about surface species. In fact, the band at 1305 cm⁻¹ is very likely due to gas phase methane. Furthermore, I think that in order to show clear indications of the presence of formate, background should be subtracted, and the relevant spectral range should be zoomed.

Response 5: We thank the referee for the comment.

1) In this study, the term “semi *in-situ*” refers to the experimental method we employed to characterize the catalyst under certain external conditions. Due to the limitations of the instrument, it was beyond our capability to conduct completely *in-situ* characterization of photothermal catalysis by XPS or FT-IR. Instead, we implemented a semi *in-situ* approach where the reactor was simultaneously illuminated and externally heated for a specified time, followed by rapid collection of the relevant XPS and FT-IR data after the removal of the light illumination. The viability of such a testing method has been validated by other researchers (see references 6-7).^{6,7}

2) We highly agree with the referee on that “**the band at 1305 cm⁻¹ is very likely due to gas phase methane**”. The observed band at 1305 cm⁻¹ can indeed be attributed to the ν(C-H) vibration of CH₄. Meanwhile, as shown in **Supplementary Fig. 26-27**, due to the high catalytic activity of Ru/MnO_x, the high intensity of the characteristic peaks of CH₄ makes it difficult to capture the peaks of the intermediates, thus limiting the information available about surface species. In order to discuss the important peak more clearly and provide a clear indication of COOH*, the enlarged spectral range has been implemented in the **Fig.3i** as suggested by the review. Meanwhile, the description has been updated in the revised manuscript as following: “**For thermocatalysis, the typical peaks of monodentate carbonates (m-CO₃²⁻, 1509 cm⁻¹) and**

v(C-H) vibration of CH₄ (1305 cm⁻¹) were apparently strengthened by increasing the reaction temperature” (Please see Page 11 in the revised manuscript)

Fig. 3i Spectra of semi *in-situ* FT-IR study of Ru/MnO_x at different conditions.

Comment 6. DFT analysis show that methane formation is favored on Ru/Mn₃O₄ with some oxygen vacancies, but these calculations should be compared with the case of MnO, as this phase is also presented in the working catalyst. In addition, If I understand correctly these calculations consider only thermal processes, and therefore the role of photonic activation is not clearly considered in these calculations.

Response 6: We appreciate the referee for the constructive comment.

1) Firstly, as the suggested by the referee, we have conducted DFT calculations on the models of Ru/MnO (200) slabs. Through comparison of the Gibbs free energy (ΔG) in rate determining step of the Ru/MnO slabs, Ru/Mn₃O₄ slabs and Ru/Mn₃O_{4-x} slabs, it is found that the Ru/Mn₃O_{4-x} have a more negative ΔG , which is conducive to CH₄ formation. Hence, in the revised manuscript, we have elaborated the description as following: “As shown in Fig. 4 and Supplementary Fig. 29-31, Ru/Mn₃O_{4-x} has a more negative Gibbs free energy (ΔG) than both Ru/Mn₃O₄ and Ru/MnO during the adsorption of CO₂, indicating a strong CO₂ adsorption capacity, which is beneficial for CO₂ hydrogenation ($\Delta G = -0.914$ eV, Ru/MnO; $\Delta G = -1.475$ eV, Ru/Mn₃O₄; $\Delta G = -1.651$ eV, Ru/Mn₃O_{4-x}). Afterwards, notable variations for the subsequent CO₂ hydrogenation were observed among Ru/MnO, Ru/Mn₃O₄ and Ru/Mn₃O_{4-x}. The formation of COOH* from CO₂* is a rate determining step (RDS) for CO₂ hydrogenation over Ru/Mn₃O_{4-x} and Ru/MnO, which requires 1.232 and 1.544 eV, respectively. The protonation and subsequent dehydration of COOH* results in the generation of the intermediate of CO*, which is the RDS for the Ru/Mn₃O₄, ($\Delta G = 1.918$ eV for Ru/Mn₃O₄). Notably, compared to HCO* formation, the CO* desorption from the catalytic surface as CO is relatively difficult for all the samples. As a result, it is favorable to yield CH₄ via further hydrogenation. It is worth mentioning that in the process of CO₂ hydrogenation, ΔG of RDS over Ru/Mn₃O_{4-x} (1.232 eV) is obviously lower than

that on Ru/Mn₃O₄ ($\Delta G= 1.918$ eV) and Ru/MnO ($\Delta G= 1.544$ eV), thus facilitating the subsequent hydrogenation steps toward CH₄.” (Please see Page 12 in the revised manuscript)

Fig. 4 Gibbs free energy pathway for the formation of HCO* and CO from CO₂ over Ru/Mn₃O₄ (321), Ru/Mn₃O_{4-x} (321) and Ru/MnO (200). The blue, red, purple, yellow, and green spheres represent the Mn, O, Ru, C, and H atoms, respectively, in the calculation model.

2) Secondly, in order to verify the role of photonic activation, we conducted DFT calculations about the densities of states (DOSs) on the models of Ru/Mn₃O_{4-x} (321) slabs that simulated dark state or light state. As shown in **Supplementary Fig. 32**, it can be seen that the conduction band that simulated light state moves to the low energy region compared with that of dark, indicating the electron density increases in light state, which is conducive to the electron transfer.^{8, 9, 10} In the revised manuscript, the description has been updated as following: “Moreover, as shown in **Supplementary Fig. 32**, compared with Ru/Mn₃O_{4-x} (321) slabs that simulated dark state, the conduction band that simulated light state moved to the low energy region, indicating that the involved photons were conducive to electron transfer, which is favorable to CO₂ hydrogenation toward CH₄.” (Please see Page 13 in the revised manuscript)

Supplementary Fig. 32 The calculated densities of states (a) and projected densities of states (b) for Ru/Mn₃O_{4-x} under dark and light conditions. Fermi levels are at 0 eV.

Reviewer #2 (Remarks to the Author):

In this manuscript, the authors synthesized a new photothermal catalyst Ru/MnO/Mn₃O₄ to convert CO₂ + H₂ into CH₄, with a superior selectivity 99.5% and a CO₂ conversion rate of 66.8%. The author characterized the structure of catalysts by XRD, XPS, FTIR, and Raman. The reaction results were analyzed under the influence of different factors such as Ru content, CO₂:H₂ ratio, temperature, light intensity, and irradiation time. They proposed some reaction pathways by DFT calculations, together with the experimental evidence from FTIR. Interestingly, they observed the Ru-mediated H-spillover effect and found the formation of COOH* is easier on Ru/Mn₃O_{4-x} by both experiment and calculation. This work is surely of interests to the community of photothermal catalytic conversion of CO₂. I would suggest the manuscript to be accepted to the journal of Nature Communications, under the condition that the authors completed the following minor revisions.

Response: We thank the referee very much for the encouraging comment and constructive suggestions. We have addressed the critical questions and concerns by the referee.

Comment 1: Could the authors provide the specific morphology and size of Ru particles on the support? For example, are they nanoparticles, atomic cluster, or single atoms?

Response 1: We thank the referee very much for the comment. Based on the referee's comment, the morphology and size of the Ru particles on the support were characterized using high angle annular dark-field scanning transmission electron microscope (HAADF-STEM). As shown in **Supplementary Fig. 2**, the obtained results clearly indicate that the Ru species were nanoclusters with an average size of 1.07 ± 0.26 nm. To address the referee's concern, in the revised manuscript, the discussion about the size of Ru particles has been supplemented: "The morphology of MnO_x did not change considerably after the addition of Ru species and the average size of the deposited Ru nanoclusters is about 1.07 ± 0.26 nm" and "The high angle annular dark-field scanning transmission electron microscope (HAADF-STEM) was operated by EM-ARM300F". (Please see Page 3 and 16 in the revised manuscript).

Supplementary Fig. 2 The high angle annular dark-field scanning transmission electron microscope (HAADF-STEM) image of the Ru/MnO_x catalyst.

Comment 2: Please clarify in the manuscript what is your definition of photo-thermal catalysis. By “photo-thermal” catalysts, do you mean traditional photocatalysis under the condition of external heating? If so, this might be an “extended” definition of photothermal catalyst, which needs to be further clarified and proved in the manuscript. For photothermal catalysts defined in the literature [see *Chem Catalysis*, 1, 52-83, 2022; *Chem Catalysis*, 1, 272-297, 2021], photothermal effect (e.g., photon energy is converted to heat) is the feature of photothermal catalysts. If the Ru/MnO/Mn₃O₄ catalyst has photothermal effect, which component (e.g., MnO_x, Ru, or both) makes the major contribution to the photothermal effect? To answer this question, the authors should show the maximum temperatures can be reached by the irradiation of Ru/MnO_x and MnO_x, respectively, with increasing irradiation time.

Response 2: We thank the referee very much for the comments again.

1) Firstly, we are very pleased to clarify that photo-thermal catalysis in the manuscript refers to photothermal co-catalysis, which was achieved by the combined effect of external heating and irradiation from the Xe lamp. As suggested by the referee, to provide a clear understanding, in the revised manuscript, we have elaborated the description as following: “In this work, we report an efficient nanostructured Ru/MnO_x catalyst composed of well-defined Ru/MnO/Mn₃O₄ for photo-thermal catalytic CO₂ hydrogenation to CH₄, which is the result of a combination of external heating and irradiation.” and “A prominent CO₂ conversion of 66.8% was achieved with a superior selectivity of 99.5% and a CH₄ production rate of 166.7 mmol g⁻¹ h⁻¹ at relatively mild temperature of 200 °C (normalized by the amount of catalyst (~ 15 mg)), which is the result of a combination of external heating and irradiation.” and “The catalytic performance of Ru/MnO_x was evaluated at 200 °C in the batch reactor setup by feeding CO₂/H₂ mixed gas (the desired temperature was achieved by a combination of external heating and irradiation from the Xe lamp) and CH₄ was identified as the dominant products, with no liquid products produced” and “Then, the external heating and the 300W UV-Xe lamp (Beijing China Education Au-Light Co., Ltd) with an intensity of 2.5 W cm⁻² were both contributed to maintain the reactor temperature at 200 °C.” (Please see Page 1, Page 3, Page 5 and Page 15 in the revised manuscript).

2) Secondly, in order to investigate the photothermal effect of the Ru/MnO_x catalyst, we conducted a series of experiments to compare the maximum temperatures reached by irradiating Ru/MnO_x and MnO_x separately, with increasing irradiation time. As shown in **Supplementary Fig. 16**, under 2.5 W cm⁻² illumination, the measured average temperature of Ru/MnO_x reached 137.9 °C, higher than that of MnO_x (115.4 °C), indicating that both Ru and MnO_x contributed to the photothermal effect. In the revised manuscript, the description has been updated as

following: “Meanwhile, due to the broadening of the wavelength range of light absorption, a strong photothermal effect was expected.^{11,12} As shown in **Supplementary Fig. 16**, under 2.5 W cm⁻² illumination, the measured average temperature of Ru/MnO_x reached 137.9 °C, higher than that of MnO_x (115.4 °C), indicating that both Ru and MnO_x contributed to the photothermal effect.” and “The temperature of samples was recorded by an infrared thermal imaging camera (Fotrpic 315, Shanghai Thermal Imaging Technology Co., Ltd.)” (Please see Page 8 and Page 17 in the revised manuscript)

Supplementary Fig. 16 Infrared thermal images captured for (a) MnO_x and (b) Ru/MnO_x under 2.5 W cm⁻² illumination.

Comment 3. In Figure 2C, how do you control the temperature of photothermal catalyst? In principle, reaching a steady (or maximum) temperature of photothermal catalyst depends on the nature of photothermal effect of materials. If external heating is used to control the photothermal temperature, how may that influence the photothermal effect? In another word, will the photothermal effect of catalysts be influenced by the condition of external heating?

Response 3: We thank the referee very much for bringing this important question to our attention.

1) Firstly, it is clarified that the temperature of 200 °C was achieved by the combined effect of external heating and irradiation from the Xe lamp. As illustrated in **Supplementary Fig. 9**, in order to ensure accurate temperature measurement and to maintain uniform temperature throughout the entire reaction system, the thermocouple was positioned at a distance of 1 cm above the catalyst, in the middle of the reactor. This was done to avoid any contact between the thermocouple and the bottom of the reactor, which could result in inaccurate temperature readings. By doing so, the temperature measurement accurately reflects the temperature of the entire catalytic reaction system. To provide a clear understanding of the experimental conditions and avoid any misleading information, in the revised manuscript, we have elaborated the description as following: “The catalytic performance of Ru/MnO_x was evaluated at 200 °C in the batch reactor setup by feeding CO₂/H₂ mixed gas (the desired temperature was achieved by a combination of external heating and irradiation from the Xe lamp) and CH₄ was identified as the

dominant product, with no liquid products produced (**Supplementary Fig. 9**)” and “Then, the external heating and the 300W UV-Xe lamp (Beijing China Education Au-Light Co., Ltd) with an intensity of 2.5 W cm^{-2} were both contributed to maintain the reactor temperature at $200 \text{ }^\circ\text{C}$.” (Please see Page 5 and Page 15 in the revised manuscript).

Supplementary Fig. 9 (a) Photograph of the apparatus setup for photo-thermal CO_2 experiments in the batch reactor; (b) Schematic illustration of the photo-thermal reactor.

2) Secondly, regarding the influence of external heating on the photothermal effect of the catalyst, we acknowledge that it is an important consideration. The photothermal effect of materials depends on their intrinsic properties and the nature of the photothermal mechanism. In our experimental configuration, although the Ru/MnO_x catalyst exhibits the photothermal effect, leading to the generation of heat during irradiation, the measured average temperature of Ru/MnO_x reached $137.9 \text{ }^\circ\text{C}$ under 2.5 W cm^{-2} illumination, which is lower than the expected set temperature (as mentioned in our response to Comment 2). Therefore, the photothermal effect of the catalyst may be affected by external heating, which serves as a regulator to maintain consistency between the catalyst temperature and the reaction system temperature.

Furthermore, we also utilized an infrared thermal camera and thermochromic temperature indicator to measure the surface and bottom temperatures of the catalyst during photothermal catalytic reactions. As recorded by an infrared thermal camera, the average temperature of catalyst surface approached $203 \text{ }^\circ\text{C}$ (**Supplementary Fig. 10a**). Additionally, as shown in **Supplementary Fig. 10b**, we employed a commercially available thermochromic temperature indicator to measure the temperature at the bottom of the catalyst, which is lower than $210 \text{ }^\circ\text{C}$. These results further validate that the external heating can effectively balance the temperature influence caused by the photothermal effect, thus maintaining consistency with the set temperature.

Supplementary Fig. 10 (a) Infrared thermal images captured for the catalyst surface temperature under 2.5 W cm^{-2} irradiation, 0.1 MPa and external heating (Set temperature: $200 \text{ }^\circ\text{C}$); (b) The temperature at the bottom of the catalyst, measured using a commercially available thermochromic temperature indicator.

Comment 4. On Line 111, which vibrational peak is the blue shift of 7 cm^{-1} relative to?

Response 4: We thank the referee very much for the critical comment. To address the referee's concern, in the revised manuscript, the blue shift of the vibration peak 7 cm^{-1} has been clarified by the following statement: “As illustrated in **Supplementary Fig. 5**, compared with the pristine MnO_x , the introduction of Ru species led to a blue shift of ~ 7 wavenumbers, and the main peak at 637 cm^{-1} is assigned to A_{1g} mode of crystalline Mn_3O_4 , validating the strong metal-support interaction between Ru and MnO_x .” (Please see Page 4 in the revised manuscript)

Comment 5. For the proposed reaction pathways in Figure 4, please comment on which step(s) can be the rate determining step (RDS) for the overall reaction? Energetically, the formation of HCO^* seems to be the RDS, the diagram is lack of transition states. Is the formation of COOH^* the RDS? If not, why do you think that the difference in COOH^* formation between $\text{Ru}/\text{Mn}_3\text{O}_4$ and $\text{Ru}/\text{Mn}_3\text{O}_{4-x}$ can cause significant difference in their overall reaction kinetics? If possible, the authors should provide the transition search results.

Response 5: We thank the referee very much for the comment. Based on the referee's comment, we conducted the search for the transition state in theoretical calculations and the results are shown in **Fig. 4**. For $\text{Ru}/\text{Mn}_3\text{O}_{4-x}$, the step of formation COOH^* is the rate-determining step of the reaction. Together with the FT-IR spectroscopic characterization above, it was rationalized that the synergy between photon energy and thermal energy favored the formation of COOH^* , thus exerting a positive impact on the CO_2 methanation over Ru/MnO_x . According to the reviewer's suggestion, in the revised manuscript, we have elaborated the description as following: “As shown in **Fig. 4 and Supplementary Fig. 29-31**, $\text{Ru}/\text{Mn}_3\text{O}_{4-x}$ has a more negative Gibbs free energy (ΔG) than both $\text{Ru}/\text{Mn}_3\text{O}_4$ and Ru/MnO during the adsorption of CO_2 , indicating a strong CO_2 adsorption capacity, which is beneficial for CO_2 hydrogenation

($\Delta G = -0.914$ eV, Ru/MnO; $\Delta G = -1.475$ eV, Ru/Mn₃O₄; $\Delta G = -1.651$ eV, Ru/Mn₃O_{4-x}). Afterwards, notable variations for the subsequent CO₂ hydrogenation were observed among Ru/MnO, Ru/Mn₃O₄ and Ru/Mn₃O_{4-x}. The formation of COOH* from CO₂* is a rate determining step (RDS) for CO₂ hydrogenation over Ru/Mn₃O_{4-x} and Ru/MnO, which requires 1.232 and 1.544 eV, respectively. The protonation and subsequent dehydration of COOH* results in the generation of the intermediate of CO*, which is the RDS for the Ru/Mn₃O₄, ($\Delta G = 1.918$ eV for Ru/Mn₃O₄). Notably, compared to HCO* formation, the CO* desorption from the catalytic surface as CO is relatively difficult for all the samples. As a result, it is favorable to yield CH₄ via further hydrogenation. It is worth mentioning that in the process of CO₂ hydrogenation, ΔG of RDS over Ru/Mn₃O_{4-x} (1.232 eV) is obviously lower than that on Ru/Mn₃O₄ ($\Delta G = 1.918$ eV) and Ru/MnO ($\Delta G = 1.544$ eV), thus facilitating the subsequent hydrogenation steps toward CH₄.” (Please see Page 12 in the revised manuscript)

Fig. 4 Gibbs free energy pathway for the formation of HCO* and CO from CO₂ over Ru/Mn₃O₄ (321), Ru/Mn₃O_{4-x} (321) and Ru/MnO (200). The blue, red, purple, yellow, and green spheres represent the Mn, O, Ru, C, and H atoms, respectively, in the calculation model.

Comment 6. Please clarify if the Ru-mediated H-spillover effect in thermal catalysis remains the same under the photothermal condition. Can the light irradiation influence the H-spillover effect, why or why not?

Response 6: We thank the referee very much for bringing this important question to our attention. Based on the referee’s comment, we have conducted a series of experiments to investigate the influence of irradiation on the H-spillover effect under photothermal conditions. Firstly, we employed WO₃ as a means to quantify the extent of H-spillover effect, by which the spillover hydrogen can migrate and readily react with yellow WO₃, resulting in a dark coloration.^{13, 14} Additionally, to ensure that the temperature induced by the photothermal effect

remains below the designated temperature, we conducted the tests at 80 °C with a light intensity of 0.3 W cm⁻² under 1 MPa H₂ (**Supplementary Fig. 23**). As depicted in **Fig. 3f**, it was revealed that when exposed to a H₂ atmosphere, the color of WO₃ remained unchanged under both photothermal and thermal conditions. In contrast, the mixture of Ru/MnO_x and WO₃ exhibited a darker color under photothermal conditions compared to pure thermal conditions. This observation suggests that under photothermal catalysis, photons irradiation can enhance the H-spillover effect, thereby promoting CO₂ hydrogenation reaction. According to the reviewer's suggestion, in the revised manuscript, we have elaborated the description as following: "In addition, to investigate the impact of photons on the H-spillover effect under photothermal conditions, we employed WO₃ as a means to quantify the extent of H-spillover effect, by which the spillover hydrogen can migrate and readily react with yellow WO₃, resulting in a dark coloration. The experiment was conducted at 80 °C with a light intensity of 0.3 W cm⁻² under 1 MPa H₂ to ensure that the temperature induced by the photothermal effect remained below the designated temperature (**Supplementary Fig. 23**). As shown in **Fig. 3f**, it was revealed that the color of WO₃ remained unchanged under both photothermal and thermal conditions. In contrast, the mixture of Ru/MnO_x and WO₃ exhibited a darker color under photothermal conditions compared to thermal conditions. This observation suggests that under photothermal catalysis, the irradiation can enhance the H-spillover effect, thereby promoting the subsequent CO₂ hydrogenation reaction." and "In a typical experiment, a mixture containing 1 g of WO₃ and 0.015 g of catalyst was placed in a quartz glass culture dish. Then the quartz glass culture dish was placed in stainless steel reactor of 180 mL (CEL-MPR, Beijing China Education Au-Light Co., Ltd.). Prior to photo-thermal reaction, the reactor was sealed and the air was replaced by H₂ for three times, followed by filling with H₂ (1 MPa). Then, the external heating and the 300W UV-Xe lamp (Beijing China Education Au-Light Co., Ltd) with an intensity of 0.3 W cm⁻² were both contributed to maintain the reactor temperature at 80 °C. After the desired reaction time, the color change of the powder samples was recorded." (Please see Page 11 and Page 17 in the revised manuscript)

Fig. 3f Photographs of WO₃ and the mixture of Ru/MnO_x and WO₃ samples after treatment with H₂ at 80 °C with a light intensity of 0.3 W cm⁻² for 20 min.

Reviewer #3 (Remarks to the Author):

In this work, Zhai and collaborators present a new catalyst based on Ru sites supported on MnO_x for the low-temperature photo-thermal methanation of CO₂. The as-prepared catalyst displayed a remarkable catalytic activity and CH₄ selectivity under reaction conditions owing to the synergy between thermal and non-thermal contributions of light. Mechanistic studies indicate a decrease in the apparent activation energy and an enhancement in the formation of COOH* intermediates under illumination, thus favoring the methanation reaction.

The production of solar fuels and chemicals using CO₂ as feedstock has raised as an interesting alternative to both tackle carbon dioxide emissions and energy crisis. In this context, photo-thermal catalysis overcomes the limitations of traditional photo-catalysis by synergistically combining thermal and non-thermal contributions of sunlight, thus becoming a very dynamic and promising field of research. The results presented by Zhai and collaborators seem reassuring, however, given the vast amount of works on photo-thermal catalysis for CO₂ methanation using Ru-based catalyst, I cannot perceive any significant advance in the field. In addition to this, the role of light and heat in the overall reaction mechanism has not been completely discussed and this can lead to misinterpretations in the overall reaction pathway. Furthermore, authors did not provide any stability test to evaluate the long-term activity of the catalyst under reaction conditions.

For these reasons, I cannot recommend the publication of this work in Nature Communications in the present form. Detailed comments to support this decision and suggestions to improve the quality of this work can be found below:

Response: We thank the referee very much. The critical comments and advices by the referee are highly helpful for us to improve the quality of the manuscript. We have further conducted extensive computational and experimental investigations to well address all the referee's concerns.

Comment 1. Authors should provide an analysis on the particle size distribution of Ru on the surface of MnO_x. From the available images it is impossible to have an idea of the size of the Ru particles.

Response 1: We thank the referee very much for the comment. Based on the referee's comment, we have conducted high angle annular dark-field scanning transmission electron microscope (HAADF-STEM) characterization to identify the size of Ru particles on the surface

of MnO_x . As shown in **Supplementary Fig. 2**, it was discovered that the average size of the deposited Ru nanoclusters was about 1.07 ± 0.26 nm. In the revised manuscript, the discussion about the size of Ru nanoclusters has been supplemented: “The morphology of MnO_x did not change considerably after the addition of Ru species and the average size of the deposited Ru nanoclusters exhibit is about 1.07 ± 0.26 nm” and “The high angle annular dark-field scanning transmission electron microscope (HAADF-STEM) was operated by EM-ARM300F”. (Please see Page 3 and 16 in the revised manuscript).

Supplementary Fig. 2 The high angle annular dark-field scanning transmission electron microscope (HAADF-STEM) image of the Ru/ MnO_x catalyst.

Comment 2. Authors report remarkable methane productions in the order of hundreds of $\text{mmol g}^{-1} \text{h}^{-1}$. Are these catalytic rates normalized by the amount of catalyst (15 mg) or the total amount of Ru present in the sample?

Response 2: We thank the referee again. In fact, the catalytic rates were normalized by the amount of catalyst (~ 15 mg). In the revised manuscript, we have clarified the description by the following statement: “A prominent CO_2 conversion of 66.8% was achieved with a superior selectivity of 99.5% and a CH_4 production rate of $166.7 \text{ mmol g}^{-1} \text{h}^{-1}$ at relatively mild temperature of $200 \text{ }^\circ\text{C}$ (normalized by the amount of catalyst (~ 15 mg)).” (Please see Page 3 in the revised manuscript)

Comment 3. When it comes to the photo-thermal experiments, could the authors explain in detail the position of the thermocouple in the setup? Is it in contact with the catalyst bed or inserted in the reactor wall? Imprecise temperature measurements can lead to misinterpretations in the contributions of photon and thermal energy to the overall catalytic performance, for instance, in the calculation of apparent activation energy.

Response 3: We thank the referee very much for bringing this important question to our attention. As suggested by the referee, a detailed explanation regarding the position of the thermocouple in the experimental setup was provided. As illustrated in **Supplementary Fig. 9**, in order to ensure accurate temperature measurement and to maintain uniform temperature

throughout the entire reaction system, the thermocouple was positioned at a distance of 1 cm above the catalyst, in the middle of the reactor. This was done to avoid any contact between the thermocouple and the bottom of the reactor, which could result in inaccurate temperature readings. By doing so, the temperature measurement can accurately reflect the temperature of the entire catalytic reaction system, while also ensuring the correct calculation of the apparent activation energy of the overall reaction. Furthermore, we also utilized an infrared thermal camera and thermochromic temperature indicator to measure the surface and bottom temperatures of the catalyst during photothermal catalytic reactions. As recorded by an infrared thermal camera, the average temperature of catalyst surface approached 203 °C (**Supplementary Fig. 10a**). Additionally, as shown in **Supplementary Fig. 10b**, we employed a commercially available thermochromic temperature indicator to measure the temperature at the bottom of the catalyst, which is lower than 210 °C. This further validates that the actual temperature closely aligns with the set temperature. To provide a clear understanding of the experimental conditions and avoid any misleading information, in the revised manuscript, we have elaborated the description as following: “The catalytic performance of Ru/MnO_x was evaluated at 200 °C in the batch reactor setup by feeding CO₂/H₂ mixed gas (the desired temperature was achieved by a combination of external heating and irradiation from the Xe lamp) and CH₄ was identified as the dominant product, with no liquid products produced (**Supplementary Fig. 9-10**)” and “Then, the external heating and the 300W UV-Xe lamp (Beijing China Education Au-Light Co., Ltd) with an intensity of 2.5 W cm⁻² were both contributed to maintain the reactor temperature at 200 °C.” (Please see Page 5 and Page 15 in the revised manuscript).

Supplementary Fig. 9 (a) Photograph of the apparatus setup for photo-thermal CO₂ experiments in the batch reactor; (b) Schematic illustration of the photo-thermal reactor.

Supplementary Fig. 10 (a) Infrared thermal images captured for the catalyst surface temperature under 2.5 W cm^{-2} irradiation, 0.1 MPa and external heating (Set temperature: $200 \text{ }^{\circ}\text{C}$); (b) The temperature at the bottom of the catalyst, measured using a commercially available thermochromic temperature indicator.

Comment 4. In Fig. 2D, authors studied the effect of the light intensity on the catalytic activity. Was the temperature constant at $200 \text{ }^{\circ}\text{C}$ throughout all the intensities? Do the authors attribute the improvement in the performance only to pure non-thermal effects? It is hard to imagine a scenario in which the temperature of the catalyst does not increase upon increasing light intensity, specially taking into account its broad light absorption across the visible and infrared.

Response 4: We thank the referee very much for bringing this important question to our attention. It is clarified that the experiments with the different light intensity were conducted at a constant temperature of $200 \text{ }^{\circ}\text{C}$ (**Fig. 2d**), which was achieved by the combined effect of external heating and irradiation from the Xe lamp (as mentioned in our response to Comment 3). Furthermore, as mentioned by the referee, considering that the Ru/MnO_x catalyst has a wide absorption across the visible and infrared, which results in a temperature increase upon photon introduction, we have conducted a series of experiments to investigate the photothermal effect of the Ru/MnO_x catalyst. As shown in **Supplementary Fig. 16**, under 2.5 W cm^{-2} illumination, the measured average temperature of Ru/MnO_x reached $137.9 \text{ }^{\circ}\text{C}$, much lower than the set temperature of $200 \text{ }^{\circ}\text{C}$, which further indicates that the heat generated by irradiation will be balanced by external heating and the improvement in performance under photothermal conditions at $200 \text{ }^{\circ}\text{C}$ is attributed to the introduction of photons. As the light intensity increases, both the charge transfer and photo-induced surface reactions were enhanced, leading to an overall enhancement in catalytic activity. In the revised manuscript, the description has been updated as following: “Meanwhile, due to the broadening of the wavelength range of light absorption, a strong photothermal effect was expected. As shown in **Supplementary Fig. 16**, under 2.5 W cm^{-2} illumination, the measured average temperature of Ru/MnO_x reached $137.9 \text{ }^{\circ}\text{C}$, higher than that of MnO_x ($115.4 \text{ }^{\circ}\text{C}$), indicating that both Ru and MnO_x contributed to the

photothermal effect.” and “The temperature of samples was recorded by an infrared thermal imaging camera (Fotrpic 315, Shanghai Thermal Imaging Technology Co., Ltd.)” (Please see Page 8 and Page 17 in the revised manuscript)

Supplementary Fig. 16 Infrared thermal images captured for (a) MnO_x and (b) Ru/MnO_x under 2.5 W cm⁻² illumination.

Comment 5. In Fig 2F, authors represented the CH₄ production as a function of the irradiation time. Why did the authors stop the experiment after 4 hours? Longer reaction times would show if higher conversions are achievable.

Response 5: We thank the referee again. On the basis of the referee’s comment, we have investigated the effect of reaction time on CH₄ evolution. It was validated that 4 h was the optimal reaction time and the catalyst displayed a decent CH₄ activity of 166.7 mmol g⁻¹ h⁻¹. By further extending the reaction time, the yield of CH₄ still increased, but the evolution rate slowed down. Based on the referee’s suggestion, the updated experimental results have been included in Fig. 2f.

Fig. 2f CH₄ evolution as a function of reaction time over Ru/MnO_x. Reaction conditions: 15 mg of catalyst, full-arc 300 W UV-xenon lamp, 2.5 W cm⁻², 200 °C, initial pressure 1 MPa (H₂/CO₂=4/1).

Comment 6. Results show a very high methane selectivity in most of the experiments assuming a total reaction time of 4 hours. What happens at shorter reaction times? Is still CH₄ the main product?

Response 6: We thank the referee very much for the helpful comment. As shown in **Fig. 2f** (as mentioned in our response to Comment 5), our experimental results demonstrate that regardless of the reaction time, the selectivity of CH₄ exceeds 99%. Such a high selectivity further confirms the advantage of Ru/MnO_x in CO₂ methanation. Based on the referee's suggestion, the updated experimental results have been included in **Fig. 2f**.

Comment 7. Authors did not provide any stability test of the catalyst, so it is not possible to assess if the material is stable upon consecutive reuses. This type of study is vital to evaluate the practical application of the catalyst, so I encourage authors to perform a series of (at least) five consecutive runs to study the catalyst recyclability.

Response 7: We thank the referee very much for the suggestion. Based on the referee's suggestion, we have tested the stability of the Ru/MnO_x catalyst. Recognizing the inherent limitations posed by water vapor accumulation in the batch reactor, which hindered the accurate assessment of the catalyst's stability, we conducted experiments in the fixed-bed reactor to further evaluate the stability of the catalyst.¹⁵ As illustrated in **Supplementary Fig. 13-14**, under the conditions of 200 °C and 2.5 W cm⁻² irradiation, the catalytic activity of Ru/MnO_x remained stable after 20 hours at a high gas hourly space velocity (GHSV) of 40000 mL g⁻¹ h⁻¹. A CO₂ conversion of 29.5% was achieved with an excellent selectivity of 99.5% and a high space time yield (STY) of 95.8 mmol_{CH₄} g⁻¹ h⁻¹. These results demonstrate the excellent stability of the catalyst.

In the revised manuscript, we have elaborated the description as following, including the experimental setup: "Furthermore, the photothermal catalytic performance of the Ru/MnO_x catalyst was also assessed in a fixed-bed reactor. As illustrated in **Supplementary Fig. 13-14**, under the conditions of 200 °C and 2.5 W cm⁻² irradiation, the catalytic activity of Ru/MnO_x remained stable after 20 hours at a high gas hourly space velocity (GHSV) of 40000 mL g⁻¹ h⁻¹. A CO₂ conversion of 29.5% was achieved with an excellent selectivity of 99.5% and a high space time yield (STY) of 95.8 mmol_{CH₄} g⁻¹ h⁻¹. These results demonstrate the excellent stability of the catalyst." and "The photothermal CO₂ conversion are also performed in the fixed-bed reactor (CEL-GPPCM, Beijing China Education Au-Light Co., Ltd.) at 200 °C. 150 mg of catalyst and CO₂/H₂ mixed flow (20 mL min⁻¹/80 mL min⁻¹) were used. A 300W UV-Xe lamp (Beijing China Education Au-Light Co., Ltd) was used as the light source for the reaction (light intensity: 2.5 W cm⁻²). The products in the effluent gas were periodically analyzed by using a gas chromatograph (GC-7920, Beijing China Education Au-Light Co., Ltd.). STY of CH₄ (mol_{CH₄} g⁻¹ h⁻¹), was calculated according to the following equation

$$\text{CH}_4 \text{ STY} = \frac{F_{\text{CO}_2, \text{in}} \times X_{\text{CO}_2} \times S_{\text{CH}_4}}{W_{\text{cat}} \times V_m}$$

where $F_{\text{CO}_2, \text{in}}$ is the volumetric flow rate of CO_2 , X_{CO_2} is the CO_2 conversion, S_{CH_4} is the CH_4 selectivity, W_{cat} is the overall mass of catalyst (g), and V_m is the ideal molar volume of CO_2 at standard temperature and pressure.” (Please see Page 7 and Page 15 in the revised manuscript).

Supplementary Fig. 13 The images of (a) the photo-thermal catalytic performance evaluation process carried out in the flow reaction system and (b) the fixed-bed quartz tube reactor.

Supplementary Fig. 14 The photothermal catalytic performance of Ru/MnO_x catalyst in a fixed-bed reactor. Reaction conditions: 150 mg of catalyst, full-arc 300 W UV-xenon lamp, 2.5 W cm^{-2} , 200°C , initial pressure 0.1 MPa, CO_2/H_2 mixed flow ($20 \text{ mL min}^{-1}/80 \text{ mL min}^{-1}$).

Comment 8. Both steady-state and time-resolved PL suggest a charge transfer from MnO_x to Ru sites under irradiation. Is this electronic transfer thermodynamically favored? Could the authors provide a band diagram showing the corresponding potentials of MnO_x and metallic Ru? Furthermore, authors did not clarify the specific role of these electrons in the overall reaction pathway.

Response 8: We thank the referee very much for the comments again.

1) Firstly, based on the referee’s suggestion, the flat band potentials of MnO_x was investigated by Mott-Schottky plots. As shown in **Supplementary Fig. 19a**, MnO_x is confirmed

as a p-type semiconductor due to the negative slope and the valence band (VB) edge potential was evaluated to be c.a. 0.29 (0.49 eV vs. NHE, $E_{\text{NHE}}=E_{\text{Ag}/\text{AgCl}} + 0.197$), while the flat-band potential is 0.1–0.3 eV lower than the valence band potential in the p-type semiconductor. In addition, the band gap can be estimated to be 1.26 eV for MnO_x (Supplementary Fig. 19b).¹⁶ Considering that the work function of Ru is 4.71 eV, the charge transfer from MnO_x to Ru sites is thermodynamically favored upon light.¹⁷ In the revised manuscript, we have clarified the description by the following statement: “Moreover, the flat band potentials of MnO_x was investigated by Mott-Schottky plots. As shown in Supplementary Fig. 19a, MnO_x is confirmed as a p-type semiconductor due to the negative slope. Meanwhile, the valence band (VB) edge potential was evaluated to be c.a. 0.29 (0.49 eV vs. NHE, $E_{\text{NHE}}=E_{\text{Ag}/\text{AgCl}} + 0.197$), while the flat-band potential is 0.1-0.3 eV lower than the valence band potential in the p-type semiconductor. In addition, the band gap can be estimated to be 1.26 eV for MnO_x (Supplementary Fig. 19b). Consequently, as shown in Supplementary Fig. 20, considering that the work function of Ru is 4.71 eV, the photo-excited electrons can facily transfer from MnO_x to Ru sites under light irradiation.” (Please see Page 8 in the revised manuscript).

Supplementary Fig. 19 (a) Mott–Schottky plots of the MnO_x ; (b) The bandgap value of the MnO_x .

Supplementary Fig. 20 The work function of Ru and band structures of MnO_x .

2) Secondly, in order to provide a comprehensive understanding of the role of electrons in the overall reaction pathway, we employed WO_3 as a means to study the influence of electrons on H-spillover effect.^{13, 14} The experiment was conducted at 80 °C with a light intensity of 0.3 W cm^{-2} under 1 MPa H_2 to ensure that the temperature induced by the photothermal effect remains below the designated temperature (**Supplementary Fig. 23**). As depicted in **Fig. 3f**, the photo reveals that when exposed to a H_2 atmosphere, the color of WO_3 remained unchanged under both photothermal and thermal conditions. In contrast, the mixture of Ru/MnO_x and WO_3 exhibited a darker color under photothermal conditions compared to thermal conditions. This observation suggests that the introduction of photo-generated carriers can enhance the H-spillover effect, thereby promoting CO_2 hydrogenation reaction. In addition, as shown in **Fig. 2e** and **Fig. 3i**, the Arrhenius plot and the FT-IR spectroscopic characterization indicate that the introduction of photo-generated carriers can reduce the activation energy of the reaction and promote the generation of COOH^* . Furthermore, as shown in **Fig. 4**, for $\text{Ru/Mn}_3\text{O}_{4-x}$, the step of formation COOH^* is recognized as the rate-determining step of the reaction. Therefore, it was rationalized that the introduction of photo-generated carriers can enhance the H-spillover effect, reduce the activation energy of the reaction and promote the generation of COOH^* , thus exerting a positive impact on the CO_2 methanation over Ru/MnO_x .

In the revised manuscript, we have elaborated the description as following: “In addition, to investigate the potential impact of photons on the H-spillover effect under photothermal conditions, we employed WO_3 as a means to quantify the extent of H-spillover effect, by which the spillover hydrogen can migrate and readily react with yellow WO_3 , resulting in a dark coloration. The experiment was conducted at 80 °C with a light intensity of 0.3 W cm^{-2} under 1 MPa H_2 to ensure that the temperature induced by the photothermal effect remained below the designated temperature (**Supplementary Fig. 23**). As shown in **Fig. 3f**, it was revealed that the color of WO_3 remained unchanged under both photothermal and thermal conditions. In contrast, the mixture of Ru/MnO_x and WO_3 exhibited a darker color under photothermal conditions compared to thermal conditions. This observation suggests that under photothermal catalysis, the irradiation can enhance the H-spillover effect, thereby promoting the subsequent CO_2 hydrogenation reaction.” and “As shown in **Fig. 4 and Supplementary Fig. 29-31**, $\text{Ru/Mn}_3\text{O}_{4-x}$ has a more negative Gibbs free energy (ΔG) than both $\text{Ru/Mn}_3\text{O}_4$ and Ru/MnO during the adsorption of CO_2 , indicating a strong CO_2 adsorption capacity, which is beneficial for CO_2 hydrogenation ($\Delta G = -0.914$ eV, Ru/MnO ; $\Delta G = -1.475$ eV, $\text{Ru/Mn}_3\text{O}_4$; $\Delta G = -1.651$ eV, $\text{Ru/Mn}_3\text{O}_{4-x}$). Afterwards, notable variations for the subsequent CO_2 hydrogenation were observed among Ru/MnO , $\text{Ru/Mn}_3\text{O}_4$ and $\text{Ru/Mn}_3\text{O}_{4-x}$. The formation of COOH^* from CO_2^* is a rate determining step (RDS) for CO_2 hydrogenation over $\text{Ru/Mn}_3\text{O}_{4-x}$ and Ru/MnO , which requires 1.232 and 1.544 eV, respectively. The protonation and subsequent dehydration of COOH^* results in the generation of the intermediate of CO^* , which is the RDS for the

Ru/Mn₃O₄, ($\Delta G = 1.918$ eV for Ru/Mn₃O₄). Notably, compared to HCO* formation, the CO* desorption from the catalytic surface as CO is relatively difficult for all the samples. As a result, it is favorable to yield CH₄ via further hydrogenation. It is worth mentioning that in the process of CO₂ hydrogenation, ΔG of RDS over Ru/Mn₃O_{4-x} (1.232 eV) is obviously lower than that on Ru/Mn₃O₄ ($\Delta G = 1.918$ eV) and Ru/MnO ($\Delta G = 1.544$ eV), thus facilitating the subsequent hydrogenation steps toward CH₄,” and “In a typical experiment, a mixture containing 1 g of WO₃ and 0.015 g of catalyst was placed in a quartz glass culture dish. Then the quartz glass culture dish was placed in stainless steel reactor of 180 mL (CEL-MPR, Beijing China Education Au-Light Co., Ltd.). Prior to photo-thermal reaction, the reactor was sealed and the air was replaced by H₂ for three times, followed by filling with H₂ (1 MPa). Then, the external heating and the 300W UV-Xe lamp (Beijing China Education Au-Light Co., Ltd) with an intensity of 0.3 W cm⁻² were both contributed to maintain the reactor temperature at 80 °C. After the desired reaction time, the color change of the powder samples was recorded.” (Please see Page 11, Page 12 and Page 17 in the revised manuscript)

Fig. 3f Photographs of WO₃ and the mixture of Ru/MnO_x and WO₃ samples after treatment with H₂ at 80 °C with a light intensity of 0.3 W cm⁻² for 20 min.

Fig. 4 Gibbs free energy pathway for the formation of HCO* and CO from CO₂ over Ru/Mn₃O₄ (321), Ru/Mn₃O_{4-x} (321) and Ru/MnO (200). The blue, red, purple, yellow, and green spheres represent the Mn, O, Ru, C, and H atoms, respectively, in the calculation model.

Comment 9. In Table S2 in SI, please include the amount of Ru in all the samples. For a fair comparison and to avoid misleading conclusions, results should clearly indicate that the methane production rate has been normalized per total mass of catalyst or total mass of Ru. TOF calculations should be also included in the table.

Response 9: We thank the referee very much for the suggestion. As suggested by the referee, we have revised **Supplementary Table 3** by including the amount of Ru in all the samples. Meanwhile, TOF calculations has also been included in the table. Of note, the CH₄ production rate was normalized per total mass of catalyst rather than per total mass of Ru. Please see the changes in the Supplementary Information.

Supplementary Table 3 The summarized CH₄ yields for recently reported photo-thermo-catalysts.

Catalysts	Metal loading (wt%)	H ₂ :CO ₂ ratio	Pressure (Mpa)	Light sources	Light intensity (W cm ⁻²)	Temperature (°C)	CH ₄ production rate (mmol g ⁻¹ h ⁻¹)	CO ₂ conversion (%)	CH ₄ selectivity (%)	TOF (h ⁻¹)	Ref
Ru/MnO _x	7.3	4:1	1	300 W Xe lamp 200-1100 nm	2.5	200 (external heater)	166.7	66.8	99.5	232	This work
Co ₇ Cu ₁ Mn ₁ O _x (200)	—	3:1	0.1	300 W Xe lamp 300-1100 nm	0.234	200 (external heater)	14.5	27.45	85.3	—	2
Ru/Al ₂ O ₃	2.4	4:1	0.08	1000 W Xe lamp	—	396	115	0.95	99.2	484	18
Cu ₂ O/Graphene		4:1	0.13	300 W Xe lamp	0.2	250 (external heater)	14.93 (Cu)	2.84	99	0.256	19
Ru@Ni ₂ V ₂ O ₇	0.35	4:1	0.067	300 W Xe lamp	2	350	114.9	96.3	99.3	3340	20
Ru/Mg(OH) ₂	11.5	1:1	0.1	300 W Xe lamp	1.8	—	44.85	1.68	69.5	56.7	21
Rh/Al	5	3:1	1.5	300 W Xe lamp	11.3	200 (external heater)	550	—	99	1132	22
21%Co/Al ₂ O ₃	0.21	4:1	0.1	300 W Xe lamp 200-1100 nm	1.3	292	6.04	—	97.7	1.74	23
Ru-TiO _x	1.77	4:1	—	300 W Xe lamp	2.0	276	22.35	—	99.99	12.76	24
Ir@UiO66	0.14	4:1	0.1	300 W Xe lamp	2.3	250 (external heater)	19.9 (Flow reactor)	9.3	95	2876	25
8 % Ru/SiO ₂	0.8	6:1	—	300 W Xe lamp	0.063	300 (external heater)	55.44 (Flow reactor)	51.8	99	70	26
Ru-Al ₂ O _{3-x} -L	0.7	4:1	0.1	300 W Xe lamp	2.27	236	0.84	86.47	99	1248	3
Ru/H _x MoO _{3-y}	4	1:1	—	300 W Xe lamp Vis-IR	0.75	140 (external heater)	20.8 (Flow reactor)	—	99	52.6	27

References

1. Jiang H, *et al.* Light-driven CO₂ methanation over Au-grafted Ce_{0.95}Ru_{0.05}O₂ solid-solution catalysts with activities approaching the thermodynamic limit. *Nature Catalysis* **6**, 519-530 (2023).
2. He Z, *et al.* Photothermal CO₂ hydrogenation to hydrocarbons over trimetallic Co–Cu–Mn catalysts. *Green Chemistry* **23**, 5775-5785 (2021).
3. Liu X, *et al.* Strong interaction over Ru/defects-rich aluminium oxide boosts photothermal CO₂ methanation via microchannel flow-type system. *Advanced Energy Materials* **12**, 2201009 (2022).
4. Tang Y, *et al.* Morphology-dependent support effect of Ru/MnO_x catalysts on CO₂ methanation. *Colloids and Surfaces A: Physicochemical and Engineering Aspects* **630**, 127636 (2021).
5. Dreyer JAH, *et al.* Influence of the oxide support reducibility on the CO₂ methanation over Ru-based catalysts. *Applied Catalysis B: Environmental* **219**, 715-726 (2017).
6. Phung-on I, Saiyasombat C. Semi in-situ local structure observation during PWHT of Cr–Mo weldments extent of the PWHT on local structure changes in heat affected zone microstructure. *Journal of Materials Research and Technology* **11**, 1123-1134 (2021).
7. Gomez MA, *et al.* Further insights into the Fe(ii) reduction of 2-line ferrihydrite: a semi in situ and in situ TEM study. *Nanoscale Advances* **2**, 4938-4950 (2020).
8. Li Y, *et al.* Experimental and theoretical insights into an enhanced CO₂ methanation mechanism over a Ru-based catalyst. *Applied Catalysis B: Environmental* **319**, 121903 (2022).
9. Yang X, *et al.* Oxygen vacancy-induced spin polarization of tungsten oxide nanowires for efficient photocatalytic reduction and immobilization of uranium(VI) under simulated solar light. *Applied Catalysis B: Environmental* **324**, 122202 (2023).
10. Li Y, *et al.* Robust photo-assisted removal of NO at room temperature: Experimental and density functional theory calculation with optical carrier. *Green Energy & Environment* **8**, 1102-1116 (2023).
11. Zhang F, *et al.* Photothermal catalytic CO₂ reduction over nanomaterials. *Chem Catalysis* **1**, 272-297 (2021).
12. Song C, Wang Z, Yin Z, Xiao D, Ma D. Principles and applications of photothermal catalysis. *Chem Catalysis* **2**, 52-83 (2022).
13. Xing S, Xiong M, Zhao S, Zhang B, Qin Y, Gao Z. Improving the efficiency of hydrogen spillover by an organic molecular decoration strategy for enhanced catalytic hydrogenation performance. *ACS Catalysis* **13**, 4003-4011 (2023).
14. Wang C, *et al.* Product Selectivity Controlled by Nanoporous Environments in Zeolite Crystals Enveloping Rhodium Nanoparticle Catalysts for CO₂ Hydrogenation. *Journal of the American Chemical Society* **141**, 8482-8488 (2019).

15. Zhu J, *et al.* Dynamic structural evolution of iron catalysts involving competitive oxidation and carburization during CO₂ hydrogenation. *Science Advances* **8**, eabm3629.
16. Zheng J, Lei Z. Incorporation of CoO nanoparticles in 3D marigold flower-like hierarchical architecture MnCo₂O₄ for highly boosting solar light photo-oxidation and reduction ability. *Applied Catalysis B: Environmental* **237**, 1-8 (2018).
17. Fu R, wang y, Wang G, Zhan Q, Zhang L, Liu L. Defect ZrO_{2-x} supported Ru nanoparticles as Mott-Schottky photocatalyst for efficient ammonia synthesis at ambient conditions. *Green Chemistry*, (2023).
18. Grote R, *et al.* Collective photothermal effect of Al₂O₃-supported spheroidal plasmonic Ru nanoparticle catalysts in the sunlight-powered Sabatier reaction. *ChemCatChem* **12**, 5618-5622 (2020).
19. Mateo D, Albero J, García H. Photoassisted methanation using Cu₂O nanoparticles supported on graphene as a photocatalyst. *Energy & Environmental Science* **10**, 2392-2400 (2017).
20. Chen Y, *et al.* Cooperative catalysis coupling photo-/photothermal effect to drive Sabatier reaction with unprecedented conversion and selectivity. *Joule* **5**, 3235-3251 (2021).
21. Kong N, *et al.* Ruthenium nanoparticles supported on Mg(OH)₂ microflowers as catalysts for photothermal carbon dioxide hydrogenation. *ACS Applied Nano Materials* **3**, 3028-3033 (2020).
22. Fu G, *et al.* Rh/Al nanoantenna photothermal catalyst for wide-spectrum solar-driven CO₂ methanation with nearly 100% selectivity. *Nano Letters* **21**, 8824-8830 (2021).
23. Chen X, *et al.* MOF-templated preparation of highly dispersed Co/Al₂O₃ composite as the photothermal catalyst with high solar-to-fuel efficiency for CO₂ methanation. *ACS Applied Materials & Interfaces* **12**, 39304-39317 (2020).
24. Dong T, *et al.* Ru decorated TiO_x nanoparticles via laser bombardment for photothermal co-catalytic CO₂ hydrogenation to methane with high selectivity. *Applied Catalysis B: Environmental* **326**, 122176 (2023).
25. Tang Y, *et al.* Encapsulating Ir nanoparticles into UiO-66 for photo-thermal catalytic CO₂ methanation under ambient pressure. *Journal of Materials Chemistry A* **10**, 12157-12167 (2022).
26. Kim C, *et al.* Energy-efficient CO₂ hydrogenation with fast response using photoexcitation of CO₂ adsorbed on metal catalysts. *Nature Communications* **9**, 3027 (2018).
27. Ge H, Kuwahara Y, Kusu K, Bian Z, Yamashita H. Ru/H_xMoO_{3-y} with plasmonic effect for boosting photothermal catalytic CO₂ methanation. *Applied Catalysis B: Environmental* **317**, 121734 (2022).

REVIEWER COMMENTS

Reviewer #1 (Remarks to the Author):

I appreciate the important efforts undertaken by the authors to clarify the points raised by in the previous version of the article. I particularly appreciate the additional analyses and calculations performed. I think these new results almost completely clarify the significant doubts I had about the previous version of this contribution. Yet, there are still a few points, that I would like to ascertain before I can recommend this communication for publication. In particular, the following aspects require further explanation:

- Despite the additional explanations, I still not very convinced of the conveniency of using the term “semi in situ” and I would suggest the authors to drop it to avoid confusion.
- Regarding assignment of the FTIR bands, the presence of gas phase methane is manifest not only by the band at 1305 cm⁻¹ but also by the one at about 3012 cm⁻¹, which is clearly visible in Figs. S26-S27. However, the presence of surface formate cannot be unambiguously establish considering just one single band, attributed to the asymmetric $\nu(\text{OCO})$ because the corresponding symmetric stretching contribution should be also visible at around at around 1374 cm⁻¹. However, details are hard to ascertain in the spectra of Fig 3I and therefore, in opinion, these FTIR does not provide much information about the mechanism, although they confirm that under illumination methane is formed at low temperature. Therefore, I would suggest the authors to reconsider the FTIR discussion.
- In Fig 2b shows that selectivity slightly decline at low H₂/CO₂ ratio. Is CO detected under those conditions as a minor product? How is the CO₂ conversion affected by the H₂/CO₂ ratio? This last aspect is of clear interest.
- What is the diameter of the quartz reactor shown in Fig S13?. My feeling is that with that setup illumination of the catalyst is not very efficient because the inside part is basically in the dark. So, can you reproduce in that continuous flow reactor the differences between thermal and photothermal conditions shown in Fig 2c? Some comments on this respect would be illuminating

Reviewer #2 (Remarks to the Author):

The authors have properly addressed all my comments and questions. The manuscript is significantly improved. I would suggest the manuscript to be accepted in the journal of Nature Communications.

Reviewer #3 (Remarks to the Author):

In their revision, Zha and collaborators have addressed most of my previous comments and have included additional discussions that certainly improved the quality of the work. In any case, before accepting for publication I would like to draw authors' attention to some points:

a) When it comes to comment 7, authors have now included a 20-h stability test. From Supplementary Figure 14, it is clear that the catalyst deactivates under reaction conditions. The drop in conversion is quite obvious, showing a distinct decreasing trend (from ~40 to 29.5 %). For this reason, authors shouldn't claim that "These results demonstrate the excellent stability of the catalyst". Indeed, it is exactly the opposite and authors should explain the reason for the deactivation of the catalyst. A post-mortem characterization of the catalyst is required to address the stability issue.

b) From both the images and discussion provided by the authors, it seems that the thermocouple is located 1 cm above the catalyst, directly exposed to the light beam. Authors explain that the 200 °C temperature is the result of the light illumination together with the external heating. How did the authors perform the dark experiments? If the thermocouple is located 1 cm above the catalyst bed and the temperature was set at 200 °C then the actual temperature at the bottom should be much higher than 200 °C.

c) Which is the thickness of the catalyst bed? It is hard to imagine a situation in which no thermal gradient is observed between the bottom and the surface temperature of the catalyst, especially at high light intensities.

Response Letter

Manuscript ID: NCOMMS-23-30118A

Title: “Photo-thermal coupling to enhance CO₂ hydrogenation toward CH₄ over Ru/MnO/Mn₃O₄”

We are very grateful to the referees for the critical comments and the constructive suggestions, which helped us to improve the quality of the manuscript. We have carefully responded to all the questions point-by-point, and have revised the manuscript thoroughly. The changes have been highlighted by yellow background in the revised manuscript.

Reviewer #1 (Remarks to the Author):

I appreciate the important efforts undertaken by the authors to clarify the points raised by in the previous version of the article. I particularly appreciate the additional analyses and calculations performed. I think these new results almost completely clarify the significant doubts I had about the previous version of this contribution. Yet, there are still a few points, that I would like to ascertain before I can recommend this communication for publication. In particular, the following aspects require further explanation:

Response: We thank the referee for the encouraging comments, and we have addressed the critical questions and concerns by the referee thoroughly.

Comment 1. Despite the additional explanations, I still not very convinced of the conveniency of using the term “semi in situ” and I would suggest the authors to drop it to avoid confusion.

Response 1: We thank the referee very much for bringing this question to our attention. Based on the referee’s comment, the term “semi *in-situ*” has been dropped in the revised manuscript.

Comment 2. Regarding assignment of the FTIR bands, the presence of gas phase methane is manifest not only by the band at 1305 cm⁻¹ but also by the one at about 3012 cm⁻¹, which is clearly visible in Figs. S26-S27. However, the presence of surface formate cannot be unambiguously establish considering just one single band, attributed to the asymmetric $\nu(\text{OCO})$ because the corresponding symmetric stretching contribution should be also visible at around at around 1374 cm⁻¹. However, details are hard to ascertain in the spectra of Fig 3I and therefore, in opinion, these FTIR does not provide much information about the mechanism, although they

confirm that under illumination methane is formed at low temperature. Therefore, I would suggest the authors to reconsider the FTIR discussion.

Response 2: We thank the referee very much for the constructive suggestions. Based on the referee's comment, we have reconsidered the FT-IR discussion. As shown in the revised **Fig.3i**, for thermocatalysis, the intermediate of formate species was observed at 1541 cm^{-1} (COOH^* , $\nu(\text{OCO})_{\text{as}}$) and 1373 cm^{-1} (COOH^* , $\nu(\text{OCO})_{\text{s}}$) when the reaction temperature increased up to $200\text{ }^\circ\text{C}$, which together confirms the presence of surface formate species. In contrast, upon light irradiation, the typical peaks of COOH^* species appeared at a lower reaction temperature of $170\text{ }^\circ\text{C}$. Hence, the involved photons were considered to be prone to accelerate the formation of intermediate COOH^* species, thus can promote the formation of CH_4 . In order to provide a clear indication of COOH^* , we have revised **Fig.3i** by highlighting the typical peak at 1373 cm^{-1} . Meanwhile, the description has been updated in the revised manuscript as following: “**Notably, the intermediate of formate species was observed at 1541 cm^{-1} (COOH^* , $\nu(\text{OCO})_{\text{as}}$) and 1373 cm^{-1} (COOH^* , $\nu(\text{OCO})_{\text{s}}$) when the reaction temperature increased up to $200\text{ }^\circ\text{C}$** ” (Please see Page 12 in the revised manuscript)

Fig.3i: Spectra of FT-IR study of Ru/MnO_x at different conditions.

Comment 3. In Fig 2b shows that selectivity slightly decline at low H_2/CO_2 ratio. Is CO detected under those conditions as a minor product? How is the CO_2 conversion affected by the H_2/CO_2 ratio? This last aspect is of clear interest.

Response 3: We thank the referee very much for bringing this important question to our attention. On the basis of the referee's comment, we have thoroughly investigated the impact of H_2/CO_2 ratio on CO_2 conversion. It was validated that CH_4 was identified as the dominant product, with a minor presence of CO and C_2H_6 . Moreover, it was observed that both the yield of CH_4 and the rate of CO_2 conversion increased with the increasing H_2 proportion in the H_2/CO_2 mixture. Notably, a distinct CH_4 production rate of $166.7\text{ mmol g}^{-1}\text{ h}^{-1}$ was obtained for Ru/MnO_x at a relatively high H_2/CO_2 ratio of 4/1, highlighting the crucial role of adequate H_2

supply during the reaction. Based on the referee's suggestion, the updated experimental results have been included in **Fig. 2b**.

Fig. 2b: Influence of CO₂/H₂ volume ratio in the feedstock on CH₄ evolution rate over Ru/MnO_x.

Comment 4. What is the diameter of the quartz reactor shown in Fig S13? My feeling is that with that setup illumination of the catalyst is not very efficient because the inside part is basically in the dark. So, can you reproduce in that continuous flow reactor the differences between thermal and photothermal conditions shown in Fig 2c? Some comments on this respect would be illuminating.

Response 4: We thank the referee again.

1) Firstly, the diameter of the quartz reactor shown in **Supplementary Fig. 13c** was 6 mm. The length, diameter and thickness of the light region of the quartz reactor were 20 mm, 6 mm and 1mm, respectively.

2) Secondly, we have conducted a series of experiments to reproduce the difference between thermal and photothermal conditions in the fixed-bed, as shown in **Fig. 2c**. As shown in **Supplementary Fig. 14**, at a gas hourly space velocity (GHSV) of 20000 mL g⁻¹ h⁻¹, the catalytic activity of Ru/MnO_x gradually increased with an increasing temperature, and its activities under photothermal conditions were higher than those under thermal conditions which further proves the involved photons were prone to promote the formation of CH₄. In the revised manuscript, we have elaborated the description as following: “Furthermore, the photothermal catalytic performance of the Ru/MnO_x catalyst was also assessed in a fixed-bed reactor. As shown in **Supplementary Fig. 13-14**, at a gas hourly space velocity (GHSV) of 20000 mL g⁻¹ h⁻¹, the catalytic activity of Ru/MnO_x gradually increased with an increasing temperature, and its activities under photothermal conditions were higher than those under thermal conditions, which further proves that the involved photons were prone to promote the formation of CH₄ in the fixed-bed reactor.” (Please see Page 7 in the revised manuscript)

Supplementary Fig. 13c Dimensions of the fixed-bed quartz tube reactor.

Supplementary Fig. 14 Temperature-dependent space time yield of CH₄ over Ru/MnO_x under photothermal (a) and thermal (b) conditions. Reaction conditions: 150 mg of catalyst, full-arc 300 W UV-xenon lamp, 2.5 W cm⁻², initial pressure 0.1 MPa, CO₂/H₂ mixture flow (10 mL min⁻¹/40 mL min⁻¹).

Reviewer #2 (Remarks to the Author):

The authors have properly addressed all my comments and questions. The manuscript is significantly improved. I would suggest the manuscript to be accepted in the journal of Nature Communications.

Response: We thank the reviewer for the comment.

Reviewer #3 (Remarks to the Author):

In their revision, Zhai and collaborators have addressed most of my previous comments and have included additional discussions that certainly improved the quality of the work. In any case, before accepting for publication I would like to draw authors' attention to some points:

Response: We thank the referee very much. We have addressed the critical concerns by the referee thoroughly.

Comment 1. When it comes to comment 7, authors have now included a 20-h stability test. From Supplementary Figure 14, it is clear that the catalyst deactivates under reaction conditions. The drop in conversion is quite obvious, showing a distinct decreasing trend (from ~40 to 29.5 %). For this reason, authors shouldn't claim that "These results demonstrate the excellent stability of the catalyst". Indeed, it is exactly the opposite and authors should explain the reason for the deactivation of the catalyst. A post-mortem characterization of the catalyst is required to address the stability issue.

Response 1: We thank the referee very much for this critical comment. Based on the referee's comment, the claim of "**These results demonstrate the excellent stability of the catalyst**" have been removed in the revised manuscript. Moreover, we have conducted a series of characterization after the reaction to explain the reason for the decreased activity. As shown in **Supplementary Fig. 16**, it was discovered that the average size of Ru nanoclusters in the used Ru/MnO_x was about 2.6 ± 0.6 nm, indicating the catalyst agglomeration. In addition, as characterized by thermogravimetric-mass spectrometry technique (TG-MS) (**Supplementary Fig. 17**), carbon deposition was observed for the used Ru/MnO_x catalyst. Therefore, catalyst agglomeration and carbon deposition were the main reasons for the deactivation of catalyst. As suggested by the referee, in the revised manuscript, the discussion has been updated as following: "Meanwhile, under the conditions of 200 °C and 2.5 W cm⁻² irradiation, the catalytic activity of Ru/MnO_x remained acceptably stable after 20 hours at a high GHSV of 40000 mL g⁻¹ h⁻¹ (**Supplementary Fig. 15**). A CO₂ conversion of 29.5% was achieved with an excellent selectivity of 99.5% and a high space time yield (STY) of 95.8 mmol_{CH₄} g⁻¹ h⁻¹. The decrease of activity was probably caused by catalyst agglomeration and carbon deposition as confirmed by the TEM and Thermogravimetric-mass spectrometric (TG-MS) of the catalyst after the reaction (**Supplementary Fig. 16-17**);" and "Thermogravimetric-mass spectrometric (TG-MS) analyses were performed on a thermogravimetric analyser (NETZSCH STA449 F3-QMS403D) instrument under air."

(Please see Page 7 and 18 in the revised manuscript).

Supplementary Fig. 16 TEM image of Ru/MnO_x after reaction of 20 h at 200 °C under photothermal condition in the fixed-bed reactor.

Supplementary Fig. 17 TG-MS analysis of Ru/MnO_x after reaction of 20 h at 200 °C under photothermal condition in the fixed-bed reactor.

Comment 2. From both the images and discussion provided by the authors, it seems that the thermocouple is located 1 cm above the catalyst, directly exposed to the light beam. Authors explain that the 200 °C temperature is the result of the light illumination together with the external heating. How did the authors perform the dark experiments? If the thermocouple is located 1 cm above the catalyst bed and the temperature was set at 200 °C then the actual temperature at the bottom should be much higher than 200 °C.

Response 2: We thank the referee again. In fact, the dark experiments were conducted at 200 °C by only external heating as measured by a thermocouple without light illumination. It is important to note that as shown in the **Supplementary Fig. 9**, the batch reactor was heated by a copper heating ring along the side wall, and the bottle of the reactor was not directly heated. Therefore, the reaction temperature can be accurately recorded by a thermocouple and the actual temperature at the bottom should not be much higher than 200 °C. Additionally, as shown in **Supplementary Fig. 10b**, we employed a commercially available thermochromic temperature indicator to measure the temperature at the bottom of the catalyst, which is lower than 210 °C.

This further validates that the actual temperature closely aligns with the set temperature. To provide a clear understanding of the experimental conditions and avoid any confusion, in the revised manuscript, we have updated **Supplementary Fig. 9**. (Please see Page 10 in the *Supplementary Information*)

Supplementary Fig. 9 (a) Photograph of the apparatus setup for photo-thermal CO₂ experiments in the batch reactor; (b) Schematic illustration of the photo-thermal reactor; (c) and (d) Schematic illustration of the heating system.

Comment 3. Which is the thickness of the catalyst bed? It is hard to imagine a situation in which no thermal gradient is observed between the bottom and the surface temperature of the catalyst, especially at high light intensities.

Response 3: We thank the referee very much for bringing this important question to our attention. Based on the comment by the referee, the thickness of the catalyst bed was measured to be $25.3 \pm 4 \mu\text{m}$ in the batch reactor (the amount of catalyst was 15 mg). Moreover, we agree with the referee that there might be thermal gradient between the bottom and the surface temperature of catalyst, which was inevitable, especially at high light intensities. Of note, due to the slim catalyst bed and the limited precision of the thermochromic temperature indicator, it proves challenging to experimentally measure these minor temperature gradients. Nonetheless, the temperature gradients might not have notable impact on the catalytic since the reaction

mainly occurred on the surface of the catalyst. Note, in our experiment, the temperature was achieved by the combined effect of external heating and irradiation from the Xe lamp.

In addition, this study focused on that the photothermal coupling to enhance CO₂ methanation. It has been clearly demonstrated by the observation that the photo-thermo-catalytic activity obviously surpasses that of the pure thermal reaction at equivalent temperatures. In the revised manuscript, we have elaborated the description regarding to the thickness of the catalyst bed as following: “The thickness of the catalyst bed was $25.3 \pm 4 \mu\text{m}$ ” and “The thickness of the catalyst bed in the batch reactor was measured by laser scanning confocal microscopy LEXT OLS5100.” (*Please see Page 15 and Page 18 in the revised manuscript*).

I hope this response can well address the referee's concern.

REVIEWERS' COMMENTS

Reviewer #1 (Remarks to the Author):

I appreciate the effort taken by the authors to clarify thae last points. I am satisfied by the answers and I think the article is now ready for publication

Reviewer #3 (Remarks to the Author):

Authors have addressed all my comments.

Response Letter

Manuscript ID: NCOMMS-23-30118B

Title: “Photo-thermal coupling to enhance CO₂ hydrogenation toward CH₄ over Ru/MnO/Mn₃O₄”

Reviewer #1 (Remarks to the Author):

I appreciate the effort taken by the authors to clarify the last points. I am satisfied by the answers and I think the article is now ready for publication

Response: We thank the reviewer for the comment.

Reviewer #3 (Remarks to the Author):

Authors have addressed all my comments.

Response: We thank the reviewer for the comment.